# $\kappa$-symmetric M5 brane web for defects in $AdS_7/CFT_6$ holography

Varun Gupta

Indian Institute of Science Education and Research,
Bhopal, Madhya Pradesh 462066, India

E-mail: varungupta@iiserb.ac.in

## Abstract

In this work, we will continue our analysis of some general probe M5 brane solutions from our previous work in $AdS_7 \times S^4$ spacetime (appeared in arxiv:2109.08551). These are codimension-2 in $AdS_7$ and preserve at least 2 supercharges when the worldvolume 3-form flux field strength is zero. We will turn on the field strength and find that the embedding conditions are modified, excluding certain branes contained in the previous result. The new main result here is very general, so we pick simpler embedding conditions that describe highly symmetric examples that preserve half of the supersymmetry of the 11 dimensions. When the flux field is zero, worldvolumes have $AdS_5 \times S^1$ topology. We turn the flux field value non-zero in these examples and analyze how the shape of the worldvolume deforms as supersymmetry is broken by some additional fractions.

# 1 Introduction

In this work, we present some new M5 probe brane solutions embedded in the background spacetime geometry of $AdS_7 \times S^4$. These probe brane solutions are dual to the codimension-2 defects in the boundary gauge theory with 6d $\mathcal{N} = (2,0)$ supersymmetry via the $AdS_7/CFT_6$ holographic correspondence. They have non-compact world volumes that extend along the radial direction of $AdS$ and end in the boundary at the locations of the non-local defects. See recent papers [5–7] on codim-2 defects using $AdS_7/CFT_6$ holography; also see [8] for a nice exposition from defect-CFT viewpoint. See [1,2] for their relation to Gukov-Witten defects [3,4] in the 4d SYM theory. In our calculation, we consider the general solutions from Section 4 of paper [9] and introduce the deformations by making the world volume self-dual 3-form flux field non-zero.

The figures 1 and 2, describe the situation from this calculation schematically. In figure 1 we depict the $AdS_7$ of the spacetime geometry as a solid cylinder with its boundary along its surface. The probe M5 brane in this figure is depicted by a surface that ends at the cylinder boundary. This probe brane ends at the location of the dual defect which is sketched in the

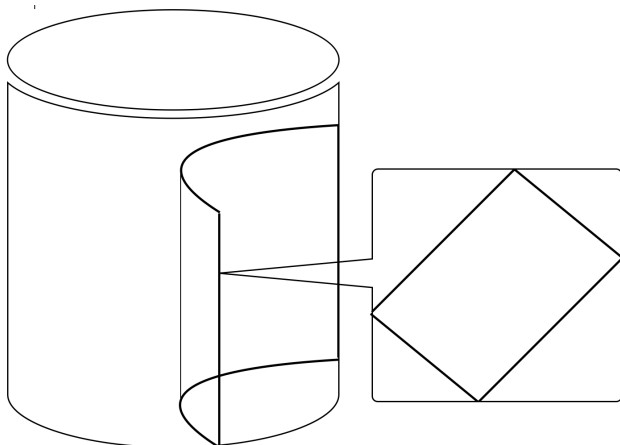

**Figure 1:** This figure depicts the probe M5 brane dual to a codimension-2 defect in the background spacetime geometry of $AdS_7 \times S^4$ that ends on the defect in the AdS boundary. The inset in this figure describes the dual defect in the gauge theory. The brane world volume carries no 3-form flux field.

inset alongside. The second schematic figure 2 estimates the situation when the 3-form flux field $h$ is turned non-zero on the worldvolume resulting in the changed shape of the probe brane as well as the dual defect.

Analysing such probe branes in the $AdS_7$ bulk is expected to reveal new features about the 4d defects in the boundary gauge theory.

For example, deforming the probe branes – due to the $AdS/CFT$ holographic correspondence – will lead to deforming the shape of the defects in the boundary theory. This in turn may also be useful for the analysis of such defect-CFTs from the bootstrap point of view where correlation functions of various operators can be determined.

For example, values of the 4-point correlation functions like

$$\langle \mathrm{D}_m(x)\mathrm{D}_n(y)X^a(w)X^b(z) \rangle_{\text{4d defect}}, \quad \langle \mathrm{D}_m(x)\mathrm{D}_n(y)\mathrm{D}_p(w)\mathrm{D}_q(z) \rangle_{\text{4d defect}}, \quad \text{etc.} \tag{1.1}$$

can be determined on the defect. Here, $\mathrm{D}_m(x)$ are called *displacement* operators of the defect, associated with the defect deformation at the location $x$. And $X^a$, $X^b$ are the scalar field operators in the 6d (2,0) tensor multiplet theory. See [10] for work on 2d Wilson surface operators and [11] for Wilson lines. But while the deformation measurement in the correlators in (1.1) includes the *quantum fluctuations*, the deformations that we analyze in this note are truly classical, obeying the BPS constraints due to $\kappa$-symmetry.

In Section 2 of this note, we show that when we break the 1/16 BPS supersymmetry of the codimension-2 brane solution from [9] by another factor of 1/2, the $\kappa$-symmetry constraint (2.6) allows us to make the flux field $h$ nonzero. Therefore, we start our calculation by considering the projection conditions given in (2.10) which break the 11d spacetime susy by a factor of 1/32 and find the modified embedding conditions in (2.25) as a main result that allow the $h$ field to take non-zero value in the $\kappa$-symmetry equation (2.6).

So this analysis tells us that making $h$ field non-zero deforms the shape of the M5 world volume breaking the susy by a factor of 1/2. We further suspect these deformations must

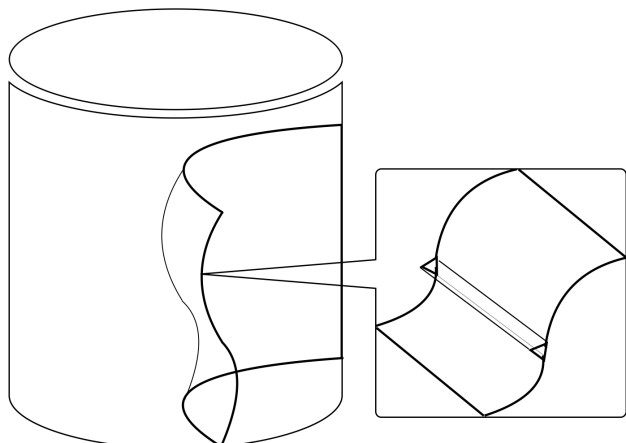

**Figure 2:** In this **second figure**, we have introduced deformations by allowing the field $h$ to take non-zero values and in our calculation, we find that embedding conditions get affected allowing for deformations under BPS constraints.

be occuring in the form of some 2d ridge-like spiked shapes emerging from the brane world volume and extending in the orthogonal directions to the brane in 11 dimensions.

In section 3 of this note, we focus on recovering some higher supersymmetric cases from the general solution given in (2.25). Since the derived solution in section 2 is very general, it is very difficult to interpret any shape associated with a brane. This solution in (2.25) is also expected to include a complicated geometry of a BPS brane-web formed out of intersections of many M5 branes. By choosing examples from the embedding condition in (2.25) which are more supersymmetric, we look for a simpler version of this equation. For example, a solution like

$$F^{(1)}(\Phi_0, \Phi_i, Z_a) = \frac{\Phi_1}{\Phi_0} = 0 \quad \text{and} \quad F^{(2)}(\Phi_0, \Phi_i, Z_a) = \frac{Z_1}{Z_2} = 0 \tag{1.2}$$

[1]belongs in this class and is a much simpler condition to express. In section 3.1, we show that this solution preserves half of the supersymmetry, including the one suggested by the projection conditions in (2.10). In our previous work [9], we have already understood that those M5 branes that are half-BPS(and without 3-form flux) have a simpler world volume shape like: $AdS_5 \times S^1$ (or $S^5 \times S^1$ for a compact worldvolume) and the corresponding defect is also of a simpler shape of $\mathbb{R} \times S^3$. In section 3, we will analyze how the world volume due to an ansatz like in (1.2) gets affected when the flux field $h$ field is turned non-zero.

## 2  New M5 solutions with flux field

The covariant action of a single probe M5 brane, first given by Pasti, Sorokin and Tonin in [12], is given below

---

[1]here $\Phi_i$s and $Z_a$s are the complexified coordinates of 11 dimensions described in appendix A.

$$S_{PST} = T_5 \int d^6 x \left( \sqrt{-\det[g_{mn} + i\widetilde{H}_{mn}]} - \frac{\sqrt{-g}}{4} \widetilde{H}^{mn} H_{mn} \right) - T_5 \int \left( \underline{A}^{(6)} - \frac{1}{2} \underline{A}^{(3)} \wedge H_3 \right)$$
(2.1)

$T_5$ is the tension of the M5 brane

$$T_5 = \frac{1}{(2\pi)^5 l_p^6} .$$
(2.2)

Here the second integral is the Wess-Zumino term where $\underline{A}^{(6)}$ and $\underline{A}^{(3)}$ are the pull-back of the 11d background gauge potential. In the action, $H_{mn}$ and $\widetilde{H}_{mn}$ are defined as

$$\widetilde{H}^{mn} = (\star H)^{mnp} v_p \qquad \left( \text{here } (\star H)_{mnp} = \frac{\varepsilon_{mnpqrs} H^{qrs}}{3! \sqrt{-g}} \right)$$

$$H^{mn} = H^{mnp} v_p$$
(2.3)

($\varepsilon$ is the Levi-Civita 6-tensor) and introduce an auxiliary field $a$ with the normalized derivative

$$v_p = \frac{\partial_p a}{\sqrt{-g^{mn} \partial_m a \partial_n a}}$$
(2.4)

The 3-form field $H_{mnp}$ above is defined as

$$H_3 = dB_2 - \underline{A}^{(3)}$$
(2.5)

where $B_2$ is a 2-form gauge potential of the 6d (2,0) tensor multiple theory on the single M5 worldvolume. In the papers [13, 14], the authors show that the M5 action in equation (2.1) is invariant under the $\kappa$-symmetry transformation. The constraint that one obtains under this fermionic symmetry is given by the equation written below

$$\Gamma_\kappa \epsilon = \pm \epsilon$$
(2.6)

where the projector is given by [15]

$$\Gamma_\kappa = \frac{1}{\sqrt{-\det g}} \left[ \gamma_{\tau 12345} + \frac{40}{6!} \varepsilon^{mnpqrs} \gamma_{mnp} h_{qrs} \right]$$
(2.7)

$\gamma_{\tau 12345}$ is the product of six worldvolume $\Gamma$ matrices. $h$ is the self-dual 3-form flux field(obeying the $h = \star_g h$ condition) and the indices $m, n, p, q, r, s$ are the 6d coordinate indices. The $h$ field here is related to the field strength $H_3$ in equation (2.5) by the equations

$$H_{mnp} = \frac{4}{Q} (\delta_m^q + k_m^q) h_{qnp}$$

$$k_m^n = h_{mpq} h^{npq}$$

$$Q = 1 - \frac{2}{3} \text{Tr} \left[ k_m^p k_p^n \right]$$
(2.8)

$\epsilon$ in the equation (2.6) is the Killing spinor of the 11-dimensional spacetime given by the expression

$$\epsilon = e^{\frac{1}{2}(\Gamma_{04}+\Gamma_1\gamma)\rho} e^{\frac{1}{2}(\Gamma_{12}+\Gamma_{45})\alpha} e^{\frac{1}{2}(\Gamma_{23}+\Gamma_{56})\beta} e^{\frac{1}{2}\Gamma_0\gamma\phi_0} e^{-\frac{1}{2}\Gamma_{14}\phi_1} e^{-\frac{1}{2}\Gamma_{25}\phi_2} e^{-\frac{1}{2}\Gamma_{36}\phi_3}$$

$$\times\, e^{\frac{1}{2}\gamma\Gamma_7\theta} e^{\frac{1}{2}\left(\Gamma_{78}+\Gamma_{9\underline{10}}\right)\chi} e^{\frac{1}{2}\Gamma_{7\underline{10}}\xi_1} e^{-\frac{1}{2}\Gamma_{89}\xi_2}\epsilon_0 \tag{2.9}$$

$\epsilon$ solves the classical Killing spinor equation of the 11d background theory whose metric solution and the choice of coordinate frame vielbein we present in appendix A of this note.

## 2.1 General solution of deformed M5 brane from $\kappa$-symmetry constraint

In this subsection, we will analyze the general M5 brane solutions from [9] that preserve the single 11d spacetime supersymmetry dictated by the projection conditions

$$\Gamma_{14}\,\epsilon_0 \,=\, \Gamma_{25}\,\epsilon_0 \,=\, \Gamma_{36}\,\epsilon_0 \,=\, -\Gamma_0\,\epsilon_0 \,=\, i\,\epsilon_0 \qquad \Gamma_{89}\epsilon_0 \,=\, -\Gamma_{7\underline{10}}\epsilon_0 \,=\, i\epsilon_0 \tag{2.10}$$

After using all the projection conditions in $\epsilon$, it takes a simplified form

$$\epsilon \,=\, e^{\frac{-i}{2}(\phi_0+\phi_1+\phi_2+\phi_3+\xi_1+\xi_2)}\left(\cos\frac{\theta}{2} \,-\, \sin\frac{\theta}{2}\,\Gamma_7\right)\epsilon_0 \tag{2.11}$$

M5 worldvolume $\gamma$ matrices are related to the 11 dimensional $\Gamma$ matrices as follows

$$\gamma_i \,=\, \mathfrak{e}_i^a\,\Gamma_a \qquad\qquad \text{index } a \text{ runs over integer values } 0,1,2,\ldots,\underline{10}, \tag{2.12}$$

here we have introduced the notation $\mathfrak{e}_i^a$ for pull-backs of the 11d frame vielbein as $\mathfrak{e}_i^a \equiv e_\mu^a\partial_i X^\mu$.

Now we consider a generic embedding of the M5 so that the 11d coordinates:

$$\phi_0(\tau,\sigma_1,\sigma_2,\sigma_3,\sigma_4,\sigma_5),\; \rho(\tau,\sigma_1,\ldots,\sigma_5),\; \ldots,\; \phi_3(\tau,\sigma_1,\ldots,\sigma_5),\; \ldots,\; \xi_2(\tau,\sigma_1,\ldots,\sigma_5) \tag{2.13}$$

which are arbitrary functions of the worldvolume coordinates $\tau,\sigma_i$s. In this calculation, we will actively use the pullbacks $\mathfrak{e}^a$s to find the generic conditions.

In our analysis, we consider the self-dual flux 3-form field $h$ to be proportional to the worldvolume forms and its expression given by

$$h = \mathcal{F}(X^m)\frac{\varepsilon^{abc}}{6}\mathfrak{e}^a \wedge \mathfrak{e}^b \wedge \mathfrak{e}^c \tag{2.14}$$

where $\mathcal{F}(X^m)$ is some functional dependence on the 11d spacetime coordinates to be determined in this section.

We are mainly going to analyze those M5s that are codimension-2 in $AdS_7$ and wrap a 1d curve on $S^4$. A subclass of solutions of such M5 world volumes was also analyzed in section 4 of reference [9]. In that work, the branes had no flux field turned on and preserved atleast 2 supersymmetries given by the projections

$$\Gamma_{14}\,\epsilon_0 \,=\, \Gamma_{25}\,\epsilon_0 \,=\, \Gamma_{36}\,\epsilon_0 \,=\, -\Gamma_{7\underline{10}}\,\epsilon_0 \,=\, i\,\epsilon_0\,. \tag{2.15}$$

Those solutions wrapped a circle along the $\xi_1$ direction on $S^4$ (with the pullback $\mathfrak{e}^9$ zero).

In another work [16], we also analyzed the branes that were codimension-4 in $AdS_7$ and wrapped an $S^3$ sphere in $S^4$ directions. Those solutions also had the world volume flux field turned on and preserved at least 2 supersymmetries with one of them given from projections in (2.10). See [17–19] for work on other similar codimension-4 branes wrapping $S^3$ in $S^4$ directions.

We will focus on the $\kappa$-symmetry constraint in equation (2.6). The calculation method we use in this section is based on the method in [20, 21] presented for a general class of $\frac{1}{16}$ BPS probe D3 brane solutions in $AdS_5 \times S^5$ carrying 5 non-zero angular momentum charges of the 10d spacetime.

For the class of solutions that we are analyzing, the following vielbein components are zero

$$\mathfrak{e}^7 = 0 \qquad \mathfrak{e}^8 = 0 \tag{2.16}$$

Therefore the 6-form $\gamma_{\tau 12345}$ will be a sum of 84 terms ($\binom{9}{6}$). Further $\varepsilon^{mnpqrs}\gamma_{mnp}h_{qrs}$ will give additional contribution in $\kappa$-symmetry constraint equation (2.6). From the first look, this analysis seems to involve an additional $84 \times 20$ 6-form terms. However, we will take the hint from [9] and proceed accordingly to find a significant reduction in this number. First, we point out that the volume form in the most general solution in [9] was given by the derived formula

$$\mathrm{dvol}_6 = \sqrt{-\det g} = \mathfrak{e}^0 \wedge \mathfrak{e}^{\underline{10}} \wedge \frac{(\omega + \tilde{\omega}) \wedge (\omega + \tilde{\omega})}{2} \tag{2.17}$$

[2]For the class of solutions in this note, this formula for the volume form becomes

$$\mathrm{dvol}_6 = \sqrt{-\det g} = \mathfrak{e}^0 \wedge \mathfrak{e}^{\underline{10}} \wedge \frac{(\mathfrak{e}^{14} + \mathfrak{e}^{25} + \mathfrak{e}^{36}) \wedge (\mathfrak{e}^{14} + \mathfrak{e}^{25} + \mathfrak{e}^{36})}{2} \tag{2.18}$$

And therefore, for the self-dual field $h$ we find that we have to take

$$h = \mathcal{F}(X^m) \left( \mathfrak{e}^0 + \mathfrak{e}^{\underline{10}} \right) \wedge \left( \mathfrak{e}^{14} + \mathfrak{e}^{25} + \mathfrak{e}^{36} \right) \tag{2.19}$$

So in the $\kappa$-symmetry equation the term $\varepsilon^{mnpqrs}\gamma_{mnp}h_{qrs}$ will be an addition of a lot less number of terms! In appendix B we collect the 6-form constraints(a result from [9]) that these solutions obey when the $h$ field is zero. From now on we will analyze how much those 6-form constraints modify when $h$ field is nonzero.

Out of the 84 terms in $\gamma_{\tau 12345}$ and many more terms in $\varepsilon^{mnpqrs}\gamma_{mnp}h_{qrs}$, we start with

$$\mathfrak{e}^{0\underline{10}}\Gamma_{0\underline{10}} \wedge \left( \mathfrak{e}^{14} \wedge \mathfrak{e}^{25}\,\Gamma_{1425} + \mathfrak{e}^{14} \wedge \mathfrak{e}^{36}\,\Gamma_{1436} + \mathfrak{e}^{25} \wedge \mathfrak{e}^{36}\,\Gamma_{2536} \right) \tag{2.20}$$

---

[2]here $\omega$ and $\tilde{\omega}$ are the pullbacks of Kähler 2-forms on $\mathbb{CP}^1$ and $\widetilde{\mathbb{CP}}^3$, respectively, defined as follows: $\omega = \mathfrak{e}^8 \wedge \mathfrak{e}^9$ and $\tilde{\omega} = \mathfrak{e}^1 \wedge \mathfrak{e}^4 + \mathfrak{e}^2 \wedge \mathfrak{e}^5 + \mathfrak{e}^3 \wedge \mathfrak{e}^6$.

+ flux terms

$$+ \mathcal{F}(X^m) \sum_{\substack{a,b,c,d,e,f \\ \in \{0,\underline{10},1,4,2,5\}}} \mathfrak{e}^{abc} \, \Gamma_{abc} \wedge \mathfrak{e}^{def} \; + \; \mathcal{F}(X^m) \sum_{\substack{a,b,c,d,e,f \\ \in \{0,\underline{10},1,4,3,6\}}} \mathfrak{e}^{abc} \, \Gamma_{abc} \wedge \mathfrak{e}^{def}$$

$$+ \, \mathcal{F}(X^m) \sum_{\substack{a,b,c,d,e,f \\ \in \{0,\underline{10},2,5,3,6\}}} \mathfrak{e}^{abc} \, \Gamma_{abc} \wedge \mathfrak{e}^{def}$$

After acting on the Killing spinor $\epsilon$ in (2.11) this combination should be equal to $\sqrt{-\det g}$ whose value is given in equation (2.18). Notice the indices $d, e, f$ in the above equation will only appear in the combinations suggested by the value of the $h$ field given in equation (2.19). So each of these summations here will only contain 4 terms. We write the expanded form below(of flux terms)

$$\mathcal{F}(X^m) \, \mathfrak{e}^{0\underline{10}} \wedge \mathfrak{e}^{14} \wedge \mathfrak{e}^{25} \left( \Gamma_{014} + \Gamma_{\underline{10}14} + \Gamma_{025} + \Gamma_{\underline{10}25} \right)$$
$$+ \mathcal{F}(X^m) \, \mathfrak{e}^{0\underline{10}} \wedge \mathfrak{e}^{14} \wedge \mathfrak{e}^{36} \left( \Gamma_{014} + \Gamma_{\underline{10}14} + \Gamma_{036} + \Gamma_{\underline{10}36} \right)$$
$$+ \mathcal{F}(X^m) \, \mathfrak{e}^{0\underline{10}} \wedge \mathfrak{e}^{25} \wedge \mathfrak{e}^{36} \left( \Gamma_{025} + \Gamma_{\underline{10}25} + \Gamma_{036} + \Gamma_{\underline{10}36} \right) \qquad (2.21)$$

After hitting the combination of (2.20) and (2.21) on $\epsilon$ in (2.11) we find that the result is equal to the volume form in (2.18) provided function $\mathcal{F}(X^m)$ is of the fixed value

$$\mathcal{F}(X^m) = \frac{1 - \sin\theta}{2\,\cos\theta} = \frac{1}{2} \frac{\cos\frac{\theta}{2} - \sin\frac{\theta}{2}}{\cos\frac{\theta}{2} + \sin\frac{\theta}{2}} \qquad (2.22)$$

This functional dependence on the 11d coordinates is the same as the value that we found in [16] where we analyzed a class of codimension-4 branes from the general solutions in [9]. This function vanishes when $\theta$ is equal to $\frac{\pi}{2}$ which is also consistent with the solutions of [9](with $h = 0$).

## Modified 6-form constraints

Next, we analyze the combinations in $\Gamma_\kappa$ where indices $a, b, c, d, e, f \in \{0, 1, 4, 2, 5, 9\}$ and $\in \{\underline{10}, 1, 4, 2, 5, 9\}$. When $h$ was zero the sum of these combinations was zero(see second equation in (B.1) for $a = 8$). In the presence of the $h$ field, we see that the 6-form constraint becomes stricter and is equal to

$$\mathfrak{e}^0 \wedge \overline{\mathbf{E}^8} \wedge \mathfrak{e}^{14} \wedge \mathfrak{e}^{25} = 0$$
$$\mathfrak{e}^{\underline{10}} \wedge \overline{\mathbf{E}^8} \wedge \mathfrak{e}^{14} \wedge \mathfrak{e}^{25} = 0 \qquad (2.23)$$

Similarly, the remaining constraints in (B.1) are also modified accordingly when appropriate combinations are considered for indices $a, b, c, d, e, f$ in the matrix $\Gamma_\kappa$

$$\mathfrak{e}^0 \wedge \overline{\mathbf{E}^a}\,\overline{\mathbf{E}^b}\,\overline{\mathbf{E}^c} \wedge \tilde{\omega} = 0 \qquad\qquad \mathfrak{e}^{\underline{10}} \wedge \overline{\mathbf{E}^a}\,\overline{\mathbf{E}^b}\,\overline{\mathbf{E}^c} \wedge \tilde{\omega} = 0$$
$$\mathfrak{e}^0 \wedge \overline{\mathbf{E}^a} \wedge \tilde{\omega} \wedge \tilde{\omega} = 0 \qquad\qquad \mathfrak{e}^{\underline{10}} \wedge \overline{\mathbf{E}^a} \wedge \tilde{\omega} \wedge \tilde{\omega} = 0 \qquad (\text{for } a, b, c = 1, 2, 3, 8)$$

$$(2.24)$$

By considering the remaining combinations in $\Gamma_\kappa$ we have checked that the other 6-form constraints are unchanged and they are the same as the one given in the appendix equations (B.4), (B.5) and (B.6).

## Embedding solution

From this analysis, we have found that the functional conditions that determine the embedding of this class of M5 solutions are given by the two arbitrary holomorphic constraints

$$F^{(I)}(\Phi_0\,,\Phi_1\,,\Phi_2\,,\Phi_3\,,Z_1,Z_2) = 0 \qquad\qquad (I = 1,2)\,, \qquad\qquad (2.25)$$

satisfying the scaling differential equations

$$\sum_{i=0}^{3} \partial_{\phi_i} F^{(I)} \;=\; 0 \qquad\qquad \sum_{i=1,2} \partial_{\xi_i} F^{(I)} \;=\; 0 \qquad\qquad (2.26)$$

along with the condition

$$\theta = \text{constant} \quad \& \quad \neq \frac{\pi}{2}\,. \qquad\qquad (2.27)$$

## Main result

Therefore from the analysis of the M5 branes that are codimension-2 in $AdS_7$, we conclude that – due to the $\mathcal{F}(X^m)$ dependence given in (2.22) – when the flux field $h$ is non-zero the brane shifts from the location of $\theta = \frac{\pi}{2}$. This solution preserves the same susy of the most general solution of [9] given by the projections in (2.10) [3] but one less susy than the solution subclass in section 4 of the same paper. The worldvolume again wraps the two $U(1)$ Hopf fibre directions in $AdS_7$ and $S^4$ generated by vector fields dual to $\mathfrak{e}^0$ and $\mathfrak{e}^{10}$, respectively. But the scaling condition has now been modified to the stricter conditions given in (2.26).

## Comments

In this analysis, we have also found that for the subclass of M5 branes analyzed in section 4 of ref. [9], supersymmetry is broken by another half when the flux field $h$ is non-zero. The 1-form pullback $\mathfrak{e}^9$ was zero for that solution subclass [4]. Without the $h$ field that subclass preserved an additional supersymmetry given by 4 projections in equation (2.15). A very similar result was obtained for the $\frac{1}{8}$ BPS dual giant D3 brane solutions in the $AdS_5 \times S^5$ spacetime in [22] where additional 1/2 susy got broken when the world volume electromagnetic fluxes were turned on. In [23] we showed that such $\frac{1}{8}$ BPS dual giant D3 branes with compact worldvolumes, that carry 3 non-zero angular momentum charges in the $S^5$ directions, belong in the same class of solutions which also admits non-compact branes holographically dual to the 2d surface defects in the 4d $\mathcal{N} = 4$ SYM theory.

---

[3]In [9] the most general solution contained both types of probe M5s that were codimension-2 and codimension-4 in $AdS_7$.

[4]Although, we haven't put $\mathfrak{e}^9$ equal to zero, explicitly in the above text, it can be checked easily that even with $\mathfrak{e}^9$ assumed to be zero, the steps presented above remain unaffected and an extra half of susy breaking due to (2.10) is still necessary.

# 3 Cases with higher supersymmetry

In the previous section, we have derived a class of general solution that carry non-zero 3-form flux field strength $h$ on their world volume. The embedding solution in (2.25) is very general and includes various cases: from a complex web-like structure made of intersecting M5 branes to *simpler* shapes like $AdS_5 \times S^1$ whose deformations can be analysed with some intuitions when the flux field is made to be non-zero. The $AdS_7/CFT_6$ holographic duals of the M5 brane examples(when $h$ is zero) discussed in this section will be defects of topology $\mathbb{R} \times S^3$. Later in this section(in subsection 3.5), we will also comment on the consequences for the dual defects when the 3-form $h$ field is given non-zero values obtained in the equation (2.19).

## 3.1 Case: $F^{(1)} = \Phi_1 = 0$; $F^{(2)} = Z_2 = 0$;

In our first example of a *simpler* shape worldvolume, for the two general holomorphic conditions in (2.25), we consider the conditions

$$F^{(1)}(\Phi_0\,,\Phi_1\,,\Phi_2\,,\Phi_3\,,Z_1, Z_2) = \Phi_1 = 0$$
$$F^{(2)}(\Phi_0\,,\Phi_1\,,\Phi_2\,,\Phi_3\,,Z_1, Z_2) = Z_2 = 0 \tag{3.1}$$

with $\theta =$ fixed value. Under the static gauge choice

$$\tau \to \phi_0,\ \sigma_1 \to \rho,\ \sigma_2 \to \beta,\ \sigma_3 \to \phi_2,\ \sigma_4 \to \phi_3,\ \sigma_5 \to -\xi_1 \tag{3.2}$$

the induced worldvolume metric is given by

$$ds^2 = -4l^2 \Big( \cosh^2 \rho \, d\phi_0^2 - d\rho^2 - \sinh^2 \rho \left( d\beta^2 + \cos^2 \beta \, d\phi_2^2 + \sin^2 \beta \, d\phi_3^2 \right) \Big) + l^2 \sin^2 \theta \, d\xi_1^2 \tag{3.3}$$

This is of topology $AdS_5 \times S^1$. The $\kappa$-symmetry constraint with no flux turned on looks like

$$\gamma_{\tau\sigma_1\sigma_2\sigma_3\sigma_4\sigma_5} \epsilon = \sqrt{-\det g}\, \epsilon\,. \tag{3.4}$$

After some cancellations, this is the same as

$$-\Gamma_{0134 6\underline{10}}\, \epsilon = \epsilon \tag{3.5}$$

With the current embedding conditions assumed $\Phi_1 = 0$, $Z_2 = 0$(or $\alpha = \frac{\pi}{2}$, $\chi = 0$), we consider writing the spinor $\epsilon$ with the following notation

$$\epsilon = e^{\frac{1}{2}(\Gamma_{04}+\Gamma_1\gamma)\rho} e^{\frac{1}{2}(\Gamma_{12}+\Gamma_{45})\frac{\pi}{2}} e^{\frac{1}{2}(\Gamma_{23}+\Gamma_{56})\beta} e^{\frac{1}{2}\Gamma_0\gamma\phi_0} e^{-\frac{1}{2}\Gamma_{14}\phi_1} e^{-\frac{1}{2}\Gamma_{25}\phi_2} e^{-\frac{1}{2}\Gamma_{36}\phi_3}$$

$$\times \left( \cos\frac{\theta}{2} - \Gamma_{89\underline{10}} \sin\frac{\theta}{2} \right) e^{\frac{1}{2}\Gamma_{7\underline{10}}\xi_1}\, \epsilon_0 \equiv M \left( \cos\frac{\theta}{2} - \Gamma_{89\underline{10}} \sin\frac{\theta}{2} \right) e^{\frac{1}{2}\Gamma_{7\underline{10}}\xi_1}\, \epsilon_0 \tag{3.6}$$

after commuting the six product matrix $\Gamma_{0134 6\underline{10}}$ through the factor $M$ of $\epsilon$ in the $\kappa$-symmetry equation (3.5) we get in the l.h.s.

$$M \left( \cos\frac{\theta}{2} + \Gamma_{89\underline{10}} \sin\frac{\theta}{2} \right) e^{-\frac{1}{2}\Gamma_{7\underline{10}}\xi_1} \Gamma_{0235 6\underline{10}}\, \epsilon_0 \tag{3.7}$$

For the solution to be supersymmetric this must be equal to the r.h.s. in (3.5). One can check that this happens to be true only when $\theta = \frac{\pi}{2}$ and the following projection condition is imposed on the constant spinor $\epsilon_0$, killing half of its components

$$\Gamma_{0235689} \, \epsilon_0 \; = \; \epsilon_0 \, . \tag{3.8}$$

Since the product of all the 11d Gamma matrices is equal to $1(\Gamma_{0123456878910} = 1)$, this is equivalent to

$$\Gamma_{1471\underline{0}} \, \epsilon_0 \; = \; \epsilon_0 \tag{3.9}$$

which is the same as the projection condition for the half-BPS solutions analyzed in [9](with $\sqrt{Z_1} \, \Phi_1 \; = \; c_0$ as the embedding condition). Hence the solution under consideration is half-BPS with projection condition of (3.8) when the 3-form flux field $h$ is zero with its $\theta$ location at $\frac{\pi}{2}$. Consistent with the half-BPS example of [9].

## Turning the 3-form field $h$ non-zero

Next, we make some of the components of the self-dual $h$ field non-zero. We take the two of the components to be

$$h_{\tau\rho\xi_1} \; = \; -4\mathrm{a}\, l^3 \sin\theta \, \cosh\rho \, , \quad h_{\beta\phi_2\phi_3} \; = \; 4\mathrm{a}\, l^3 \sinh^3\rho \, \sin 2\beta \tag{3.10}$$

here 'a' is some arbitrary constant. The two components follow the self-duality: $h_{\tau\rho\xi_1} = \star_g \, h_{\beta\phi_2\phi_3}$. The $\kappa$-symmetry constraint looks like this

$$\frac{1}{\sqrt{-\det g}} \left[ \gamma_{\tau\rho\beta\phi_2\phi_3\xi_1} \; + \; \gamma_{\tau\rho\xi_1} h_{\beta\phi_2\phi_3} \; - \; \gamma_{\beta\phi_2\phi_3} h_{\tau\rho\xi_1} \right] \epsilon \; = \; \epsilon \tag{3.11}$$

After some algebra this simplifies to

$$\left[ -\Gamma_{01346\underline{10}} \; - \; \mathrm{a} \cosh\rho \left( \Gamma_{011\underline{0}} - \Gamma_{346} \right) \; + \; \mathrm{a} \sinh\rho \left( \Gamma_{141\underline{0}} - \Gamma_{036} \right) \right] \epsilon \; = \; \epsilon \tag{3.12}$$

and after commuting all the 3-product $\Gamma$ matrices through the exponential factor $M$ in $\epsilon$ we get following in the l.h.s.(after the factor $M$)

$$-\mathrm{a} \cos\beta \cosh\rho \, M_{\phi_0\phi_2} \left[ M_{\xi_1}^{-1} \left( \Gamma_{021\underline{0}} \cos\frac{\theta}{2} + \Gamma_{3568\underline{9}10} \sin\frac{\theta}{2} \right) - M_{\xi_1} \left( \Gamma_{356} \cos\frac{\theta}{2} + \Gamma_{0289} \sin\frac{\theta}{2} \right) \right] \epsilon_0$$

$$-\mathrm{a} \sin\beta \cosh\rho \, M_{\phi_0\phi_3} \left[ M_{\xi_1}^{-1} \left( \Gamma_{031\underline{0}} \cos\frac{\theta}{2} + \Gamma_{2568\underline{9}10} \sin\frac{\theta}{2} \right) + M_{\xi_1} \left( \Gamma_{256} \cos\frac{\theta}{2} - \Gamma_{0389} \sin\frac{\theta}{2} \right) \right] \epsilon_0$$

$$-M_{\xi_1}^{-1} \left[ \Gamma_{02351\underline{0}} \cos\frac{\theta}{2} - \mathrm{a} \sinh\rho \left( \Gamma_{011\underline{0}} \gamma \cos\frac{\theta}{2} - \Gamma_{235681\underline{9}10} \gamma \sin\frac{\theta}{2} \right) \right] \epsilon_0$$

$$+M_{\xi_1} \left[ \Gamma_{0235689} \sin\frac{\theta}{2} - \mathrm{a} \sinh\rho \left( \Gamma_{2356} \gamma \cos\frac{\theta}{2} - \Gamma_{089} \gamma \sin\frac{\theta}{2} \right) \right] \epsilon_0 \, . \tag{3.13}$$

Here $M_{\xi_1}$, $M_{\phi_0\phi_2}$, $M_{\phi_0\phi_3}$ are the exponential factors: $e^{\frac{\Gamma_{71\underline{0}}}{2}\xi_1}$, $e^{-\Gamma_0\gamma\phi_0} e^{\Gamma_{25}\phi_2}$, $e^{-\Gamma_0\gamma\phi_0} e^{\Gamma_{36}\phi_3}$, respectively. With the help of some $\Gamma$-matrix algebra and the projection condition in (3.8) it can be shown that this is equal to $\epsilon$ (in the r.h.s.), only for $\theta = \frac{\pi}{2}$ value.

This means turning on these flux components does not break the supersymmetry for the probe M5 solution; it remains half-BPS and hence its world volume would not undergo any deformation in the target 11d superspace. But this also requires the brane to be fixed at the $\theta = \frac{\pi}{2}$ location.

## 3.2    Deforming the world volume by turning on the new fluxes

Next in this subsection, we will turn on those components of the flux field $h$ suggested by the analysis in the previous section 2.1 (whose values were obtained in equation (2.19)). These flux components break some supersymmetry, giving us insight into how the shape of the brane deforms when this happens. We take the 3-form flux field to be

$$h = \left( \mathfrak{e}^0 \wedge \mathfrak{e}^1 \wedge \mathfrak{e}^4 + \mathfrak{e}^3 \wedge \mathfrak{e}^6 \wedge \mathfrak{e}^{\underline{10}} \right) \mathcal{F} . \tag{3.14}$$

The $\kappa$-symmetry constraint equation is

$$\frac{1}{\sqrt{-\det g}} \left[ \gamma_{\phi_0 \rho \beta \phi_2 \phi_3 \xi} + \left( \mathfrak{e}^{0134 6 \underline{10}} \right)_{\phi_0 \rho \beta \phi_2 \phi_3 \xi} \left( \Gamma_{014} + \Gamma_{36\underline{10}} \right) \mathcal{F} \right] \epsilon = \epsilon . \tag{3.15}$$

After some simplification, this becomes

$$\left[ - \Gamma_{0134\underline{10}} + \left( \Gamma_{014} + \Gamma_{36\underline{10}} \right) \mathcal{F} \right] M \left( \cos \frac{\theta}{2} - \Gamma_{89\underline{10}} \sin \frac{\theta}{2} \right) M_{\xi_1} \epsilon_0 = \epsilon . \tag{3.16}$$

After commuting all the $\Gamma$ matrices in the l.h.s. through the exponential factor $M$, we get the following

$$M_{\xi_1} \Gamma_{0235689} \sin \frac{\theta}{2} \epsilon_0 - M_{\xi_1}^{-1} \Gamma_{02356\underline{10}} \cos \frac{\theta}{2} \epsilon_0$$

$$+ M_{\xi_1} \left[ \cos \frac{\theta}{2} \left( \Gamma_{025} \cos^2 \beta + \Gamma_{036} \sin^2 \beta \right) \mathcal{F} - \sin \frac{\theta}{2} \Gamma_{89} \left( \Gamma_{36} \cos^2 \beta + \Gamma_{25} \sin^2 \beta \right) \mathcal{F} \right] \epsilon_0$$

$$+ M_{\xi_1}^{-1} \left[ \sin \frac{\theta}{2} \Gamma_{89\underline{10}} \left( \Gamma_{025} \cos^2 \beta + \Gamma_{036} \sin^2 \beta \right) \mathcal{F} + \cos \frac{\theta}{2} \Gamma_{\underline{10}} \left( \Gamma_{36} \cos^2 \beta + \Gamma_{25} \sin^2 \beta \right) \mathcal{F} \right] \epsilon_0$$

$$+ M_{\phi_2 \phi_3} \cos \beta \sin \beta M_{\xi_1} \left[ \sin \frac{\theta}{2} \Gamma_{89} \left( \Gamma_{26} + \Gamma_{35} \right) + \cos \frac{\theta}{2} \left( \Gamma_{026} + \Gamma_{035} \right) \right] \mathcal{F} \epsilon_0$$

$$+ M_{\phi_2 \phi_3} \cos \beta \sin \beta M_{\xi_1}^{-1} \left[ \sin \frac{\theta}{2} \Gamma_{89\underline{10}} \left( \Gamma_{026} + \Gamma_{035} \right) - \cos \frac{\theta}{2} \left( \Gamma_{26\underline{10}} + \Gamma_{35\underline{10}} \right) \right] \mathcal{F} \epsilon_0 . \tag{3.17}$$

To show that this is equal to the r.h.s. in (3.16), the terms with the exponential factor $M_{\phi_2 \phi_3}$ must vanish. So we focus on these terms first. After using the projection condition from (3.8) and setting

$$\mathcal{F} = \frac{\cos \frac{\theta}{2} - \sin \frac{\theta}{2}}{\cos \frac{\theta}{2} + \sin \frac{\theta}{2}} , \tag{3.18}$$

these terms can be written as

$$M_{\phi_2 \phi_3} \cos \beta \sin \beta \left( \cos \frac{\theta}{2} - \sin \frac{\theta}{2} \right) \left[ M_{\xi_1} \left( \Gamma_{026} + \Gamma_{035} \right) - M_{\xi_1}^{-1} \left( \Gamma_{26\underline{10}} + \Gamma_{35\underline{10}} \right) \right] \epsilon_0 . \tag{3.19}$$

These will not vanish unless we impose another projection condition on $\epsilon$, breaking the susy by another half

$$(1 + \Gamma_{2536})\, \epsilon_0 \,=\, 0\,. \tag{3.20}$$

After taking care of $M_{\phi_2 \phi_3}$ terms, we go to the remaining set of terms in the l.h.s. we wrote in (3.17). We use the two independent projections from (3.8) and (3.20) to write the l.h.s. equal to

$$M_{\xi_1} \left[ \sin\frac{\theta}{2} + \left( \cos\frac{\theta}{2} - \sin\frac{\theta}{2} \right) \Gamma_{025} \right] \epsilon_0 - M_{\xi_1}^{-1} \Gamma_{89\underline{10}} \left[ \cos\frac{\theta}{2} - \left( \cos\frac{\theta}{2} - \sin\frac{\theta}{2} \right) \Gamma_{025} \right] \epsilon_0 \tag{3.21}$$

and this is not equal to the r.h.s. in (3.16)

$$\left( M_{\xi_1} \cos\frac{\theta}{2} \,-\, M_{\xi_1}^{-1} \Gamma_{89\underline{10}} \sin\frac{\theta}{2} \right) \epsilon_0 \tag{3.22}$$

unless we impose a third projection condition on $\epsilon_0$ breaking the susy to $\frac{1}{8}$ BPS

$$\Gamma_{025}\, \epsilon_0 \,=\, \epsilon_0 \tag{3.23}$$

| | $\phi_0$ | $\phi_1$ | $\phi_2$ | $\phi_3$ | $\alpha$ | $\beta$ | $\rho$ | $\theta$ | $\chi$ | $\xi_1$ | $\xi_2$ |
|---|---|---|---|---|---|---|---|---|---|---|---|
| M5$_N$ | × | × | × | × | × | × | − | − | − | − | − |
| M5$_{\Phi_1=0;h=0}$ | × | − | × | × | − | × | × | − | − | × | − |
| M5$_{\Phi_1=0;h\neq0}$ | × | × | × | | × | | × | − | − | × | − |

**Table 1:** Table showing the intersection directions of worldvolume of $\Phi_1 = 0$ solution with the $AdS_7$ boundary when fluxes are turned off and turned on, indicating the deformations when $h = \left( \mathfrak{e}^{014} + \mathfrak{e}^{25\underline{10}} \right) \mathcal{F}$ value discussed in this subsection.

### 3.2.1 Consequence of susy breaking and analysis of the deformations

From the $\kappa$-symmetry constraint calculation for the solution ansatz: $\Phi_1 = 0$; $Z_2 = 0$; we see that if we want to place the probe brane at an arbitrary value of $\theta$ coordinate, the flux field $h$ must be proportional to the factor $\mathcal{F} = \frac{\cos\frac{\theta}{2} - \sin\frac{\theta}{2}}{\cos\frac{\theta}{2} + \sin\frac{\theta}{2}}$. This makes $h$ field equal to zero at $\theta = \frac{\pi}{2}$ and nonzero at any other location. When the flux field is equal to the one taken in (3.14) the supersymmetry breaks to $\frac{1}{8}$ due to the 3 independent projections given below

$$- \Gamma_0\, \epsilon_0 \,=\, \Gamma_{25}\, \epsilon_0 \,=\, \Gamma_{36}\, \epsilon_0 \,=\, \Gamma_{89}\, \epsilon_0 \tag{3.24}$$

This analysis also tells us that it is impossible to have the solution ansatz: $\Phi_1 = 0$; $Z_2 = 0$; as a half-BPS probe away from the coordinate location $\theta = \frac{\pi}{2}$. The supersymmetry have to be broken by at least another $\frac{1}{4}$ factor. And the brane solution is always a deformed one at arbitrary $\theta$ location.

The nature or shape of the deformation can be analyzed a bit if we look at the projection conditions given in (3.24). One can realize that if we consider a common *intersection* of our

current solution with two of the solutions that we obtained in the past in [16], given by the embedding conditions

$$M5^1_{\text{codim4}} : \qquad \Phi_1 = 0 \, ; \ \Phi_3 = 0 \, ;$$
$$M5^2_{\text{codim4}} : \qquad \Phi_1 = 0 \, ; \ \Phi_2 = 0 \, ; \tag{3.25}$$

the brane web made out of the intersection of these three will have the same susy due to the projections in (3.24).

In principle, in this brane web, the world volume of the three constituting M5 branes need not have to intersect with each other along a particular location or a common submanifold. This web can be formed when each of the three component M5 brane worldvolumes start to develop some 2-dimensional ridge-like spike deformations emerging out in the orthogonal directions. Then those spikes join with each other at certain locations away from their respective M5 worldvolume to form a complex web-like geometry preserving the 1/8 BPS supersymmetry we see from the projection conditions in (3.24). See [25,26] for the work done on BPS configurations with such multiple orthogonal spikes emerging and joining between parallel M5 branes. These spikes from the point of view of the gauge theory living on the worldvolume of the probe M5 brane can be seen as some string-like $\frac{1}{4}$ BPS configurations, in which one or two of the scalars of the 6d $\mathcal{N} = 2, 0$ tensor multiplet develop singularities at the locations of those strings and source the 3-form flux components for field $h$ that we switched on in equation (3.14).

The emergence of this picture can be further strengthened when we turn on the relevant flux field components, separately, on the world volume of the two half-BPS M5 solutions in (3.25). These solutions were derived in [16]. Both the solutions had a world volume topology of $AdS_3 \times S^3$. They were holographic duals of codimension-4 defects in the boundary gauge theory. And when we turn on the 3-form flux field on their world volume to be equal to

$$h = \left( \mathfrak{e}^0 \wedge \mathfrak{e}^8 \wedge \mathfrak{e}^9 + \mathfrak{e}^1 \wedge \mathfrak{e}^4 \wedge \mathfrak{e}^{\underline{10}} \right) \mathcal{F} \, . \tag{3.26}$$

We see that supersymmetry has to be broken for both of them in a particular way. The susy has to be broken by another factor of 1/4. For the $M5^1_{\text{codim4}}$ solution in (3.25) the deformed brane configuration will be 1/8 BPS due to the three projection conditions

$$M5^1_{\text{codim4}} : \qquad \Gamma_{25} \, \epsilon_0 = \Gamma_{89} \, \epsilon_0 = -\Gamma_{7\underline{10}} \, \epsilon_0 = -\Gamma_0 \, \epsilon_0 \, . \tag{3.27}$$

Whereas for the solution $M5^2_{\text{codim4}}$, the deformed brane configuration will be 1/8 BPS due to the projection conditions

$$M5^2_{\text{codim4}} : \qquad \Gamma_{36} \, \epsilon_0 = \Gamma_{89} \, \epsilon_0 = -\Gamma_{7\underline{10}} \, \epsilon_0 = -\Gamma_0 \, \epsilon_0 \, . \tag{3.28}$$

In conclusion, our analysis in this subsection with the flux field $h$ taken in (3.14) suggests that the solution $\Phi_1 = 0 \, ; \ Z_2 = 0 \, ;$ may exist in an M5 brane web configuration with its worldvolume deforming and developing 2d ridge-like spikes at certain locations from where $h$ field is sourced and then these spikes stretching in the directions transverse to the worldvolume and intersecting with spikes coming out from the branes of (3.25). A clearer picture of this will require a careful analysis of the 1/4 BPS configurations of the $\mathcal{N} = (2, 0)$ field theory that lives on the single probe M5. We will need to study the scalar field

profiles near the *endstrings* that source the $h$ field at the base of these M2-spikes to get an accurate picture of the web; for example, see [24, 26]. Looking at [25] for a supersymmetry analysis of a network of multiple *endstrings* of M2-spikes stretched between parallel M5 branes may also be useful. Apart from this, perhaps one can also analyze the $\kappa$-symmetry of the ridge-like spike deformations, which are interpreted as open M2 branes ending on the probe M5 worldvolume. $\kappa$ symmetry constraint for such incidence was obtained in this paper [28]. The anomaly-free action for similar open-supermembranes was also discussed in these papers [29, 30]. We will address these questions soon in future work.

### 3.2.2 Making other components of the flux field non-zero

In the remainder of this subsection, we will turn on some other components of the flux field $h$ nonzero and consider

$$ h = \left( \mathfrak{e}^0 \wedge \mathfrak{e}^3 \wedge \mathfrak{e}^6 + \mathfrak{e}^1 \wedge \mathfrak{e}^4 \wedge \mathfrak{e}^{\underline{10}} \right) \mathcal{F} \tag{3.29} $$

In $\kappa$-symmetry analysis of this case, we will find that the supersymmetry has to broken further by another factor of $1/2$. So the M5 brane configuration with this value of flux field is $1/16$ BPS and the projections are given in the equation below

$$ -\Gamma_0 \, \epsilon_0 = \Gamma_{25} \, \epsilon_0 = \Gamma_{36} \, \epsilon_0 = \Gamma_{89} \, \epsilon_0 = -\Gamma_{7\underline{10}} \, \epsilon_0 \tag{3.30} $$

We have put $\kappa$-symmetry constraint analysis of this case in the appendix C. Here we would like to compare the $1/16$ susy with the susy preserved in the other cases discussed in later subsections 3.3.1 and 3.3.2. One can see that this $\frac{1}{16}$ BPS susy is the same as the susy preserved by the solutions of subsections 3.3.1 and 3.3.2 when some particular components of the $h$ flux field are non-zero(given in equation (3.36) and (3.42), respectively). This means that after turning on these other components of flux field $h$ in (3.29), the brane solution develops spikes in the other orthogonal directions and those spikes can join the similar spikes coming from the solutions of 3.3.1 and 3.3.2(with same nonzero components of the $h$ flux field). And these deformed M5 branes with their spikes joined form a *complex* web-like geometry that preserves the $1/16$ susy of the 11d background theory given by the projections in equation (3.30).

## 3.3 Other case examples

In this subsection, we would like to analyze other *simpler* M5 brane ansatz from the general solution in (2.25) from section 2.1 and list their preserved supersymmetry. All of the probe M5 brane cases discussed in this subsection will have the worldvolume of topology $AdS_5 \times S^1$ when the fluxes are zero, and we will see how the supersymmetries are broken when the fluxes are turned on; i.e. when the brane is moved from the coordinate position $\theta = \frac{\pi}{2}$. Upon turning on the fluxes, all the brane examples considered here will become part of a bigger web of M5 branes with the web preserving the common supersymmetry of the individual constituent brane(with fluxes nonzero).

### 3.3.1  Case $F^{(1)} = \Phi_2 = 0$; $F^{(2)} = Z_2 = 0$;

In this case example, the brane solution is again half-BPS when the flux field value is 0 and fixed at the location $\theta = \frac{\pi}{2}$. The projection condition for which is given below

$$\Gamma_{0134\underline{6}\,\underline{10}}\,\epsilon_0 = \epsilon_0 \tag{3.31}$$

This is the same as the supersymmetry preserved by the half BPS brane in the previous work [9]

$$\sqrt{Z_1}\,\Phi_2 = c_0 \tag{3.32}$$

When we make the flux field non-zero by moving it away from $\theta = \frac{\pi}{2}$ and take

$$h = \left(\mathfrak{e}^{014} + \mathfrak{e}^{25\underline{10}}\right)\mathcal{F} \tag{3.33}$$

the supersymmetry needs to be broken by another factor of $\frac{1}{4}$. This makes this brane with the above flux field value $\frac{1}{8}$ BPS due to the impositions of the projection conditions[5]

$$-\Gamma_0\,\epsilon_0 = \Gamma_{14}\,\epsilon_0 = \Gamma_{36}\,\epsilon_0 = \Gamma_{89}\,\epsilon_0 \tag{3.34}$$

**Remark**: This 1/8-BPS brane configuration with the flux field (3.33) has all 4 of its supersymmetries in common with the common supersymmetries of the two half-BPS solutions found in our previous work [16]. The embedding conditions of those are given as follows

$$\begin{aligned}
M5^1_{\text{codim4}} : \quad & \Phi_2 = 0;\ \Phi_3 = 0;\\
M5^2_{\text{codim4}} : \quad & \Phi_1 = 0;\ \Phi_2 = 0;
\end{aligned} \tag{3.35}$$

Further one can show that if we turn on some of the other components of the flux field $h$ and take it to be equal to

$$h = \left(\mathfrak{e}^{025} + \mathfrak{e}^{14\underline{10}}\right)\mathcal{F} \tag{3.36}$$

the supersymmetry breaks by another factor of half to $\frac{1}{16}$ BPS given by the projections

$$-\Gamma_0\,\epsilon_0 = \Gamma_{14}\,\epsilon_0 = \Gamma_{36}\,\epsilon_0 = \Gamma_{89}\,\epsilon_0 = -\Gamma_{7\underline{10}}\,\epsilon_0 \tag{3.37}$$

### 3.3.2  Case $F^{(1)} = \Phi_3 = 0$; $F^{(2)} = Z_2 = 0$;

In this case example, the brane solution is again half-BPS when the flux field value is 0 and fixed at the location $\theta = \frac{\pi}{2}$. The projection condition for which is given below

$$\Gamma_{0124\underline{5}\,\underline{10}}\,\epsilon_0 = \epsilon_0 \tag{3.38}$$

This is the same as the supersymmetry preserved by the half BPS brane in the previous work [9]

$$\sqrt{Z_1}\,\Phi_3 = c_0 \tag{3.39}$$

When we make the flux field non-zero by moving it away from $\theta = \frac{\pi}{2}$ and take

$$h = \left(\mathfrak{e}^{014} + \mathfrak{e}^{25\underline{10}}\right)\mathcal{F} \tag{3.40}$$

---

[5]We collect the detailed step-by-step calculation with this flux field value in the appendix D.

| Solutions | $h$ field values | no. of susy | required projections |
|---|---|---|---|
| $F^{(1)} = \Phi_1 = 0;$ <br> $F^{(2)} = Z_2 = 0;$ <br> $(\,\mathfrak{e}^2 = 0\,)$ | $h = 0$ | $\frac{1}{2}$ BPS | $\Gamma_{14}\,\epsilon_0 = -\Gamma_{7\underline{10}}\,\epsilon_0$ |
| | $h = (1+\star_g)\,\sqrt{g_{\beta\beta}g_{\phi_2\phi_2}g_{\phi_3\phi_3}}$ | | |
| | $h = (\mathfrak{e}^{014} + \mathfrak{e}^{36\underline{10}})\,\mathcal{F}$ | $\frac{1}{8}$ BPS | $\Gamma_{14}\,\epsilon_0 = -\Gamma_{7\underline{10}}\,\epsilon_0$ <br> $\Gamma_{25}\epsilon_0 = \Gamma_{36}\epsilon_0 = \Gamma_{89}\epsilon_0$ |
| | $h = (\mathfrak{e}^{036} + \mathfrak{e}^{14\underline{10}})\,\mathcal{F}$ | $\frac{1}{16}$ BPS | $\Gamma_{14}\epsilon_0 = \Gamma_{89}\epsilon_0 = -\Gamma_{7\underline{10}}\epsilon_0$ <br> $\Gamma_{25}\epsilon_0 = \Gamma_{36}\epsilon_0 = \Gamma_{89}\epsilon_0$ |
| $F^{(1)} = \Phi_2 = 0;$ <br> $F^{(2)} = Z_2 = 0;$ <br> $(\,\mathfrak{e}^3 = 0\,)$ | $h = 0$ | $\frac{1}{2}$ BPS | $\Gamma_{25}\,\epsilon_0 = -\Gamma_{7\underline{10}}\,\epsilon_0$ |
| | $h = (\mathfrak{e}^{014} + \mathfrak{e}^{25\underline{10}})\,\mathcal{F}$ | $\frac{1}{8}$ BPS | $\Gamma_{25}\,\epsilon_0 = -\Gamma_{7\underline{10}}\,\epsilon_0$ <br> $\Gamma_{14}\epsilon_0 = \Gamma_{36}\epsilon_0 = \Gamma_{89}\epsilon_0$ |
| | $h = (\mathfrak{e}^{025} + \mathfrak{e}^{14\underline{10}})\,\mathcal{F}$ | $\frac{1}{16}$ BPS | $\Gamma_{25}\epsilon_0 = \Gamma_{89}\epsilon_0 = -\Gamma_{7\underline{10}}\epsilon_0$ <br> $\Gamma_{14}\epsilon_0 = \Gamma_{36}\epsilon_0 = \Gamma_{89}\epsilon_0$ |
| $F^{(1)} = \Phi_3 = 0;$ <br> $F^{(2)} = Z_2 = 0;$ <br> $(\,\mathfrak{e}^3 = 0\,)$ | $h = 0$ | $\frac{1}{2}$ BPS | $\Gamma_{36}\,\epsilon_0 = -\Gamma_{7\underline{10}}\,\epsilon_0$ |
| | $h = (\mathfrak{e}^{014} + \mathfrak{e}^{25\underline{10}})\,\mathcal{F}$ | $\frac{1}{8}$ BPS | $\Gamma_{36}\,\epsilon_0 = -\Gamma_{7\underline{10}}\,\epsilon_0$ <br> $\Gamma_{14}\epsilon_0 = \Gamma_{25}\epsilon_0 = \Gamma_{89}\epsilon_0$ |
| | $h = (\mathfrak{e}^{025} + \mathfrak{e}^{14\underline{10}})\,\mathcal{F}$ | $\frac{1}{16}$ BPS | $\Gamma_{36}\epsilon_0 = \Gamma_{89}\epsilon_0 = -\Gamma_{7\underline{10}}\epsilon_0$ <br> $\Gamma_{14}\epsilon_0 = \Gamma_{25}\epsilon_0 = \Gamma_{89}\epsilon_0$ |
| $F^{(1)}\,(\Phi_i, Z_a) = 0;$ <br> $F^{(2)} = Z_2 = 0;$ | $h = 0$ | $\frac{1}{16}$ BPS | $\Gamma_{14}\epsilon_0 = -\Gamma_{7\underline{10}}\epsilon_0 = i\,\epsilon_0$ <br> $\Gamma_{14}\epsilon_0 = \Gamma_{25}\epsilon_0 = \Gamma_{36}\epsilon_0$ |
| | $h = \#\,\mathcal{F}$ <br> (as given in eqn (2.19)) | $\frac{1}{32}$ BPS | $\Gamma_{14}\epsilon_0 = -\Gamma_{7\underline{10}}\epsilon_0 = i\,\epsilon_0$ <br> $\Gamma_{14}\epsilon_0 = \Gamma_{25}\epsilon_0 = \Gamma_{36}\epsilon_0$ <br> $\Gamma_{14}\epsilon_0 = \Gamma_{89}\epsilon_0$ |
| $F^{(1)}\,(\Phi_i, Z_a) = 0;$ <br> $F^{(2)}\,(\Phi_i, Z_a) = 0;$ | h = 0 | $\frac{1}{32}$ BPS | $\Gamma_{14}\epsilon_0 = -\Gamma_{7\underline{10}}\epsilon_0 = i\,\epsilon_0$ |
| | $h = \#\,\mathcal{F}$ <br> (as given in eqn (2.19)) | | $\Gamma_{14}\epsilon_0 = \Gamma_{25}\epsilon_0 = \Gamma_{36}\epsilon_0$ <br> $\Gamma_{14}\epsilon_0 = \Gamma_{89}\epsilon_0$ |

**Table 2:** We summarise the results for all three solution examples presented in this section, considering various values for the flux field $h$ and the supersymmetry preserved. To do the comparison, in the last two sets of rows we also write the results of the general solution obtained in section 2.1.

the supersymmetry needs to be broken by another factor of $\frac{1}{4}$. This makes this brane with the above flux field value $\frac{1}{8}$ BPS due to the impositions of the projection conditions

$$- \Gamma_0 \, \epsilon_0 \; = \; \Gamma_{14} \, \epsilon_0 \; = \; \Gamma_{25} \, \epsilon_0 \; = \; \Gamma_{89} \, \epsilon_0 \tag{3.41}$$

Further, if we change the flux field and make some other components non-zero to consider

$$h \; = \; \left( \mathfrak{e}^{025} + \mathfrak{e}^{14\underline{10}} \right) \mathcal{F} \tag{3.42}$$

the supersymmetry breaks by another factor of half to $\frac{1}{16}$ BPS given by the projections

$$- \Gamma_0 \, \epsilon_0 \; = \; \Gamma_{14} \, \epsilon_0 \; = \; \Gamma_{36} \, \epsilon_0 \; = \; \Gamma_{89} \, \epsilon_0 \; = \; -\Gamma_{7\underline{10}} \, \epsilon_0 \tag{3.43}$$

## 3.4 Onshell action value

In this subsection, we will discuss the on-shell value of the actions for the examples we saw in this section. The values for all the case examples here are the same, which was expected since all of the $AdS_5 \times S^1$ world volumes undergo similar deformations when the $h$ field is considered non-zero. All of these probe branes end in the $AdS_7$ boundary, at some 4-dimensional submanifold $\mathbb{R} \times \Sigma_3$ where $\Sigma_3$ denotes some compact space-like 3-surface.

We focus on the example $\Phi_1 = 0$; $Z_2 = 0$; we compute the Lagrangian given by PST in equation (2.1) onshell.

We choose the gauge fixing condition where we set the scalar '$a$' to be

$$a \; = \; \xi_1 \tag{3.44}$$

For the on-shell embedding solution

$$\alpha \; = \; \frac{\pi}{2} \,, \; \theta \; = \; \text{constant} \,, \; \chi \; = \; 0 \,. \tag{3.45}$$

when we consider taking the flux field $h$ to be

$$h \; = \; \left( \mathfrak{e}^{014} + \mathfrak{e}^{36\underline{10}} \right) \mathcal{F} \tag{3.46}$$

The Lagrangian density in (2.1) evaluates to the value

$$\mathcal{L}_{PST} \Big|_{\text{on-shell}} \; = \; (16 \, l^6) \, T_5 \, \sin 2\beta \, \cosh \rho \, \sinh^3 \rho \, f(\theta) \tag{3.47}$$

which can be written as a total derivative term,

$$\mathcal{L}_{PST} \Big|_{\text{on-shell}} \; = \; \left( \frac{N^2}{8\pi^3} \right) f(\theta) \, \partial_\rho \left( \sin 2\beta \, \sinh^4 \rho \right) \tag{3.48}$$

here $f(\theta)$ is some function of the fixed coordinate $\theta$, which is a constant for the solution under consideration. The action from $\mathcal{L}_{PST}$ in (3.47) is divergent. It can be regularized by using a boundary term at $\rho = \rho_\infty$. We saw that $\mathcal{L}_{PST}$ takes the similar onshell value and can be written as a total derivative as in (3.48), for the other two case examples as well. And so after adding the appropriate boundary term, we find that the action value is zero for all the examples we discussed in this section

$$S_{PST} \Big|_{\text{reg.}} \; = \; 0 \,. \tag{3.49}$$

## 3.5 Taking $AdS_7$ boundary limit and recovering the dual defects

From these probe M5 configurations, we can also learn about the holographic dual defects in boundary gauge theory. For instance, in section 4 of reference [9], for a certain subclass of embedding conditions that preserved at least 2 supersymmetry, we were able to determine the nature of coupling of the dual codimension-2 defects to the field contents of the boundary theory by taking the large value limit for $AdS$ radial coordinate: $r = 2l \sinh \rho \to \infty$.

In [9], we were able to determine how a complexified scalar in the boundary theory becomes singular at defect locations in the boundary theory. A case example that we considered in [9] was given by the embedding conditions

$$Z_1^{\frac{1}{2}} \Phi_1 = c_0; \ Z_2 = 0; \tag{3.50}$$

Upon taking the large $r$ limit in this condition, it tells us where the probe ends in the $AdS_7$ boundary, which is also the location of the dual defect in the boundary gauge theory. The projection of this probe solution on the $AdS$ boundary also gives us the topology of the defect, which is $\mathbb{R} \times S^3$ for this example. And it also tells us about the behaviour of a complexified scalar in the boundary theory

$$\lambda = \frac{c_0}{\cos \alpha \, e^{i\phi_1}} \tag{3.51}$$

where $\lambda = r \, e^{i\frac{\xi_1}{2}}$ which is identified with a complex scalar field in the boundary theory. Here, $c_0$ is a complex parameter that tells how the defect couples with the ambient gauge theory. Our first example $\Phi_1 = 0; \ Z_2 = 0;$ analyzed in this section belongs to the case example of [9] if we consider the parameter $c_0$ value to be zero. This means for this example the corresponding dual defect in the boundary gauge theory doesn't couple to the complex scalar field in the usual way(as discussed in [3, 23, 31]). However, we must still be able to determine how the dual defect couples to the other bosonic fields in the non-abelian $\mathcal{N} = 2, 0$ tensor multiplet theory. Since the embedding conditions

$$\Phi_1 = 0; \ Z_2 = 0; \tag{3.52}$$

are not the only equations that describe this probe M5 brane example. There is also a non-zero-valued flux field $h$ associated with it (the one which was given in equation (3.10)).

We expect that the dual codimension-2 defect here will couple to the bosonic fields of the boundary theory in the same way as the *rigid* surface defects of the 4d $\mathcal{N} = 4$ SYM theory do in reference [4] [6]. The *rigid* defects of [4] were the non-abelian solutions of the same BPS equations of Kapustin and Witten in [32] that were also solved by the abelian surface defects of [3]. For *rigid* surface defects, the complex scalar field and the gauge field of the boundary gauge theory attain profiles that are weaker than the $\frac{1}{x}$ pole singularity near the defect locations(it is $\frac{1}{\log x}$ singularity). Also see the recent papers [33, 34], where the half-BPS rigid defects are analysed and their $AdS/CFT$ holographic dual D3 branes are also discussed. In contrast to the *non-rigid* defects of [3], which preserve the bosonic $SO(2,2) \times SO(4)_R \times SO(2)$

---

[6]Codimension-2 defects of 6d $\mathcal{N} = 2, 0$ SCFT map to surface defects of 4d SYM theory upon doing compactification on a suitable 2d Reimann surface [1, 2].

subgroup of the $SO(2,4) \times SO(6)_R$ symmetry(of the 4d SYM theory), the *rigid* defects preserve the enhanced symmetry of $SO(2,2) \times SO(4)_R \times SO(2) \times SO(2)_R$ subgroup.

Our example in (3.52) will also map to the dual defect of such an enhanced symmetry, in comparison to the defect coming from the brane in (3.50), where $c_0$ is non-zero. Due to the $c_0 = 0$ value, there is no coupling between the two circular directions $\phi_1$ and $\xi_1$, so the defect coming from (3.52) will get to preserve an enhanced symmetry due to an extra $SO(2)_R$ factor. The defect dual to the probe solution in (3.52) will preserve the bosonic symmetry of $SO(2,4) \times SO(3)_R \times SO(2) \times SO(2)_R$ subgroup from the 6d superconformal symmetry.

In the example in subsection 3.1, we have also seen that the non-zero value of $h$ field in (3.14) causes the brane world volume to break the supersymmetry and, as a result, the space-time symmetry to get broken down to a smaller isometry subgroup, suggesting the deformation in the shape of the brane. In the probe M5 brane example here, we have 5 directions orthogonal to its worldvolume; 3 orthogonal directions in $S^4$ and 2 orthogonal directions in $AdS_7$; the orientation and the stretching of the spiked deformations in orthogonal directions could determine the change in coupling of the dual defect with bosonic fields in the boundary gauge theory. After taking the large radius limit $r \to \infty$ here (with $h \neq 0$), we will get a holographic dual codimension-2 defect whose shape is now deformed from $\mathbb{R} \times S^3$ topology to some $\mathbb{R} \times \Sigma^3$ topology. With the $S^3$ deforming in such a way that there are spikes coming out of $S^3$ and elongating in the two spacelike directions orthogonal to the defect in the 6d theory.

However, we first need a careful study of the deformation of the M5 worldvolume to get a precise picture here. The analysis of the worldvolume gauge theory will be useful in this regard. The quarter BPS configurations of the Abelian worldvolume theory with non-trivial scalar profiles will capture these spike-shaped deformations of the worldvolume in a precise manner. The understanding of these spike-shaped deformations in the $AdS_7$ boundary limit($r \to \infty$) should also become useful for determining the change in the coupling of the dual defect with other bosonic fields in the boundary gauge theory: the other three real scalar fields and the 2-form gauge potential field $B_{mn}$. While the deformations of its world volume along $AdS_7$ directions would affect the coupling of the dual defect with the potential field $B_{mn}$; world volume deformations along $S^4$ directions would affect the coupling with the scalar fields. It would also be useful to understand the quantity: displacement operator $\mathrm{D}_m(x)$ associated with these codimension-2 defects [35].

# 4 Summary and conclusion

In this work, we considered some probe M5 brane solutions from a general class of solutions derived in the previous work [9]. The solutions subclass considered here are codimension-2 in the $AdS_7$ directions and preserve at least 2 supersymmetries of 11d due to projections in (2.15) when there are no fluxes present on the world volume. In section 2.1, we showed that to turn on the fluxes for these solutions, the supersymmetry has to be broken by another factor of $1/2$. We found that the 3-form flux field $h$ should always need to be proportional to the functional factor of

$$\mathcal{F} = \frac{\cos\frac{\theta}{2} - \sin\frac{\theta}{2}}{\cos\frac{\theta}{2} + \sin\frac{\theta}{2}}. \tag{4.1}$$

This makes the flux field zero whenever the coordinate $\theta$ is equal to $\frac{\pi}{2}$. The general embedding conditions of [9] are also modified and become more constrained. The main result of this section is the following embedding conditions in terms of two arbitrary holomorphic functions satisfying some scaling conditions

$$F^{(I)}(\Phi_0\,,\Phi_1\,,\Phi_2\,,\Phi_3\,,Z_1,Z_2) = 0 \qquad\qquad (I\,=\,1,2)$$

$$\sum_{i=0}^{3} \partial_{\phi_i} F^{(I)} \;=\; 0 \qquad\qquad \sum_{i=1,2} \partial_{\xi_i} F^{(I)} \;=\; 0 \qquad\qquad (4.2)$$

Since the above solution is very general, it is very difficult to analyze how the shape of the brane gets deformed when the flux field is turned on. In section 3, we consider examples of the brane solutions that have the world volume of shape $AdS_5 \times S^1$ when the flux field $h$ is zero. Then we turned $h$ non-zero suggested by our calculations in section 2.1. We found that when the flux field is equal to

$$h \,=\, \left(\mathfrak{e}^{014} \,+\, \mathfrak{e}^{36\underline{10}}\right)\mathcal{F} \qquad\qquad (4.3)$$

the supersymmetry was broken by another factor of $1/4$. This implies that the brane world-volume must be getting deformed by developing some 2d ridge-like spikes in one or two transverse directions. The *endlines* of these spikes on the worldvolume should be sourcing the flux field $h$. We also gave evidence that for the case example: $\Phi_1 = 0; Z_1 = 0$; and $h$ field with this value; and with susy broken by $1/8$ factor, the world volume may form a complex web with some other solutions we found in [16], given in equation (3.25). This web is made out of the spikes that stretch between the three different types of M5 branes. Further when we turned on some different components of the $h$ field non-zero; $h \,=\, \left(\mathfrak{e}^{036} \,+\, \mathfrak{e}^{14\underline{10}}\right)\mathcal{F}$, we found that the supersymmetry breaks to $1/16$ factor and the $\kappa$-symmetry analysis suggests that this brane example may become a part of a larger web structure with the examples of subsections 3.3.1 and 3.3.2 also becoming a part.

In the future, we would like to understand these spiked deformations from the viewpoint of the abelian gauge theory that lives on the probe M5 world volume. These deformations should be captured by the scalar profiles of the BPS configurations in the gauge theory in a precise manner [24–26]. In $AdS$ boundary limit this can also tell us how the codimension-2 holographic dual defect couples to the field content of the non-abelian $\mathcal{N} = 2,0$ gauge theory that exists in the boundary. While the worldvolume deformations along the $S^4$ may determine the coupling of the dual defect with 3 of the real scalar of the boundary gauge theory, the deformations along the $AdS_7$ may determine the coupling with the 2-form potential field $B_{mn}(x)$. It will also be worthwhile to understand precisely how the probe with flux field value in equation (3.10) would map to the half-BPS rigid defects of references [4], [33,34].

In future, following the method given in [7,10,11,27], we would also look to treat scalar profiles -transverse to the brane world volume- around our original solution of $AdS_5 \times S^1$ worldvolume(when the new fluxes were zero) as small fluctuations and then calculate the M5 brane action. It will be interesting to compare this answer with the calculated on-shell value of M5 brane action for the deformed BPS configurations given in equation (3.49) in section 3(when the fluxes were non-zero). The calculated action from the two different

ways mentioned above will give the effective action associated with the 4d defect that exists in the boundary theory. While the second action has the classical answer; the first action calculation will also include the quantum corrections for the defect with the shape of $\mathbb{R} \times S^3$.

This could also be taken to the next level. Since the deformed worldvolume configurations that we analyzed here are classical, by again following the method of [7, 10, 11, 27], we can do the calculation of the effective action with 1-loop correction by allowing fluctuations in the values of the embedding fields $X^m(\tau, \sigma_i)$ and 3-form $h$ around the field solutions we presented in section 3.

**Acknowledgements:** The author would like to thank Sujay Ashok for helpful discussions and his valuable feedback on earlier drafts of the manuscript. The author would also like to thank Nemani V. Suryanarayana for some helpful discussions. The author is thankful to Nabamita Banerjee for the valuable support. This work is supported by the Science and Engineering Research Board(SERB) project grant with the serial number: SERB/PHY/23-24/65. The author is also grateful to the Institute of Mathematical Sciences, Chennai, for hospitality during a visit during the progress of this work.

# A   Appendix: Details of the 11-dimensional geometry

We consider the following metric of the eleven-dimensional $AdS_7 \times S^4$ geometry in global coordinates system

$$ds^2_{AdS} = - \left( 1 + \frac{r^2}{4l^2} \right) dt^2 + \frac{dr^2}{\left( 1 + \frac{r^2}{4l^2} \right)} + r^2 d\Omega_5 \tag{A.1}$$

with $d\Omega_5 = d\alpha^2 + \cos^2 \alpha \, d\phi_1^2 + \sin^2 \alpha \left( d\beta^2 + \cos^2 \beta \, d\phi_2^2 + \sin^2 \beta \, d\phi_3^2 \right)$

$$ds^2_{S^4} = l^2 \left( d\theta^2 + \sin^2 \theta (d\chi^2 + \cos^2 \chi \, d\xi_1^2 + \sin^2 \chi \, d\xi_2^2) \right) \tag{A.2}$$

The global $AdS_7$ coordinates above can be written in terms of the following complex coordinates in $\mathbb{C}^{1,3}$

$$\Phi_0 = l \cosh \rho \, e^{i\phi_0} \quad \Phi_1 = l \sinh \rho \cos \alpha \, e^{i\phi_1} \quad \Phi_2 = l \sinh \rho \sin \alpha \cos \beta \, e^{i\phi_2} \quad \Phi_3 = l \sinh \rho \sin \alpha \sin \beta \, e^{i\phi_3} \tag{A.3}$$

which define the $AdS_7$ part as the following locus in $\mathbb{C}^{1,3}$

$$-|\Phi_0|^2 + |\Phi_1|^2 + |\Phi_2|^2 + |\Phi_3|^2 = -l^2 \tag{A.4}$$

For the $S^3 \subset S^4$ we define the complex cooordinates describing it embedded in $\mathbb{C}^2$ space

$$Z_1 = \cos \chi \, e^{i\xi_1} \qquad\qquad Z_2 = \sin \chi \, e^{i\xi_2} \, . \tag{A.5}$$

The frame vielbein that we use are the following

$$e^0 = 2l \left( \cosh^2 \rho \, d\phi_0 - \sinh^2 \rho \left( \cos^2 \alpha \, d\phi_1 + \sin^2 \alpha \cos^2 \beta \, d\phi_2 + \sin^2 \alpha \sin^2 \beta \, d\phi_3 \right) \right)$$

$$e^1 = 2l \, d\rho, \quad e^2 = 2l \sinh \rho \, d\alpha, \quad e^3 = 2l \sinh \rho \sin \alpha \, d\beta$$

$$e^4 = 2l \cosh \rho \sinh \rho \left( \cos^2 \alpha \, d\phi_{01} + \sin^2 \alpha \cos^2 \beta \, d\phi_{02} + \sin^2 \alpha \sin^2 \beta \, d\phi_{03} \right)$$

$$e^5 = 2l \sinh \rho \cos \alpha \sin \alpha \left( \cos^2 \beta \, d\phi_{02} + \sin^2 \beta \, d\phi_{03} - d\phi_{01} \right)$$

$$e^6 = 2l \sinh \rho \sin \alpha \cos \beta \sin \beta \left( d\phi_{03} - d\phi_{02} \right) \tag{A.6}$$

where $r = 2l \sinh \rho$, $\phi_0 = \frac{t}{2l}$, and

$$e^7 = l\, d\theta, \quad e^8 = l \sin\theta d\chi, \quad e^9 = l \sin\theta \cos\chi \sin\chi \left( d\xi_1 - d\xi_2 \right),$$
$$e^{\underline{10}} = l \sin\theta \left( \cos^2\chi\, d\xi_1 + \sin^2\chi\, d\xi_2 \right). \tag{A.7}$$

With the above choice of frame vielbein, it becomes apparent that the $AdS_7$ part can be expressed as a $U(1)$ Hopf fibration over a Kähler manifold $\widetilde{\mathbb{CP}}^3$. Here $\widetilde{\mathbb{CP}}^3$ is a complex projective space which is identified with the space of rays passing through the origin in the complex space $\mathbb{C}^{1,3}$. Similarly for the $S^3 \subset S^4$ part, the frame vielbein is chosen so that $U(1)$ Hopf fibration over a Kähler manifold $\mathbb{CP}^1$ becomes manifest.

# B    6-form constraints when $h$ field is zero

From the terms in the first group, we have the following set of 6-form constraints

$$\left( \mathfrak{e}^0 + \mathfrak{e}^{\underline{10}} \right) \wedge \overline{\mathbf{E}^a}\, \overline{\mathbf{E}^b}\, \overline{\mathbf{E}^c} \wedge \tilde{\omega} = 0$$
$$\left( \mathfrak{e}^0 + \mathfrak{e}^{\underline{10}} \right) \wedge \overline{\mathbf{E}^a} \wedge \tilde{\omega} \wedge \tilde{\omega} = 0 \qquad \text{for } a,b,c = 1,2,3,8 \tag{B.1}$$

with the definition

$$\mathbf{E}^1 = \mathfrak{e}^1 - i\, \mathfrak{e}^4 \qquad\qquad \mathbf{E}^2 = \mathfrak{e}^2 - i\, \mathfrak{e}^5$$
$$\mathbf{E}^3 = \mathfrak{e}^3 - i\, \mathfrak{e}^6 \qquad\qquad \mathbf{E}^8 = \mathfrak{e}^8 - i\, \mathfrak{e}^9, \tag{B.2}$$

here we have also defined a real 2-form:

$$\tilde{\omega} = \mathfrak{e}^{14} + \mathfrak{e}^{25} + \mathfrak{e}^{36} = \frac{i}{2} \left( \overline{\mathbf{E}^1}\, \mathbf{E}^1 + \overline{\mathbf{E}^2}\, \mathbf{E}^2 + \overline{\mathbf{E}^3}\, \mathbf{E}^3 \right) \equiv \omega_{\widetilde{\mathbb{CP}}^3} \tag{B.3}$$

This 2-form is the pull-back of certain Kähler forms onto the worldvolume of the brane. This Kähler form is of the base manifold $\widetilde{\mathbb{CP}}^3$ when the $AdS_7$ is written as Hopf-fibration over $\widetilde{\mathbb{CP}}^3$.

The terms with a factor $\mathfrak{e}^{0\underline{10}}$ give the constraints

$$\mathfrak{e}^0 \wedge \mathfrak{e}^{\underline{10}} \wedge \overline{\mathbf{E}^a}\, \overline{\mathbf{E}^b} \wedge \tilde{\omega} = 0$$
$$\mathfrak{e}^0 \wedge \mathfrak{e}^{\underline{10}} \wedge \overline{\mathbf{E}^1}\, \overline{\mathbf{E}^2}\, \overline{\mathbf{E}^3}\, \overline{\mathbf{E}^8} = 0. \tag{B.4}$$

The BPS differential 6-form constraints from the remaining set of terms are

$$\overline{\mathbf{E}^a}\, \overline{\mathbf{E}^b} \wedge \tilde{\omega} \wedge \tilde{\omega} = 0 \qquad\qquad \text{for } a,b = 1,2,3,8. \tag{B.5}$$

The final constraint is the following:

$$\mathfrak{e}^{14} \wedge \mathfrak{e}^{25} \wedge \mathfrak{e}^{36} = 0 \tag{B.6}$$

# C  $\kappa$ symmetry analysis with $h_{036}$, $h_{14\underline{10}}$ components

In this appendix section, we turn on some other components of the flux field $h$ for the case ansatz: $\Phi_1 = 0$; $Z_2 = 0$ discussed in section 3.1. The flux field has the following components nonzero

$$\left( \mathfrak{e}^{036} + \mathfrak{e}^{14\underline{10}} \right) \mathcal{F} \tag{C.1}$$

After some simplifications $\kappa$-symmetry constraint equation takes this look

$$\left[ - \Gamma_{0134 6\underline{10}} + (\Gamma_{036} + \Gamma_{14\underline{10}}) \mathcal{F} \right] M \left( \cos\frac{\theta}{2} - \Gamma_{89\underline{10}} \sin\frac{\theta}{2} \right) M_{\xi_1} \epsilon_0 = \epsilon \tag{C.2}$$

After commuting all the six-product and 3-product $\Gamma$ matrices through the exponential factor $M$ in the killing spinor the l.h.s. in the above equation (C.2) becomes

$$M M_{\xi_1} \cos\frac{\theta}{2} \left( \cosh^2\rho \left( \cos^2\beta\, \Gamma_{036} + \sin^2\beta\, \Gamma_{025} \right) - \sinh^2\rho\, \Gamma_{2356}\, \gamma \right) \mathcal{F} \epsilon_0$$

$$+ M M_{\xi_1} \sin\frac{\theta}{2} \left( \Gamma_{0235689} - \cosh^2\rho\, \Gamma_{89} \left( \cos^2\beta\, \Gamma_{25} + \sin^2\beta\, \Gamma_{36} \right) \mathcal{F} - \sinh^2\rho\, \Gamma_{089}\, \gamma\, \mathcal{F} \right) \epsilon_0$$

$$+ M M_{\xi_1}^{-1} \cos\frac{\theta}{2} \left( - \Gamma_{0235 6\underline{10}} + \cosh^2\rho \left( \cos^2\beta\, \Gamma_{25\underline{10}} + \sin^2\beta\, \Gamma_{36\underline{10}} \right) \mathcal{F} - \sinh^2\rho\, \Gamma_{0\underline{10}}\, \gamma\, \mathcal{F} \right) \epsilon_0$$

$$+ M M_{\xi_1}^{-1} \sin\frac{\theta}{2} \Gamma_{89\underline{10}} \left( \cosh^2\rho \left( \cos^2\beta\, \Gamma_{036} + \sin^2\beta\, \Gamma_{025} \right) - \sinh^2\rho\, \Gamma_{2356}\, \gamma \right) \mathcal{F} \epsilon_0$$

$$- M M_{\phi_2\phi_3} M_{\xi_1} \cosh^2\rho \cos\beta \sin\beta \left[ \cos\frac{\theta}{2} \left( \Gamma_{026} + \Gamma_{035} \right) + \sin\frac{\theta}{2} \Gamma_{89} \left( \Gamma_{26} + \Gamma_{35} \right) \right] \mathcal{F} \epsilon_0$$

$$+ M M_{\phi_2\phi_3} M_{\xi_1}^{-1} \cosh^2\rho \cos\beta \sin\beta \left[ \cos\frac{\theta}{2} \left( \Gamma_{26\underline{10}} + \Gamma_{35\underline{10}} \right) - \sin\frac{\theta}{2} \Gamma_{89\underline{10}} \left( \Gamma_{026} + \Gamma_{035} \right) \right] \mathcal{F} \epsilon_0$$

$$+ M M_{\phi_0\phi_2} M_{\xi_1} \cosh\rho \sinh\rho \cos\beta \left[ \cos\frac{\theta}{2} \left( \Gamma_{356} + \Gamma_{0236}\, \gamma \right) - \sin\frac{\theta}{2} \Gamma_{89} \left( \Gamma_{02} + \Gamma_5\, \gamma \right) \right] \mathcal{F} \epsilon_0$$

$$+ M M_{\phi_0\phi_2} M_{\xi_1}^{-1} \cosh\rho \sinh\rho \cos\beta \left[ \cos\frac{\theta}{2} \left( \Gamma_{02\underline{10}} + \Gamma_{5\underline{10}}\gamma \right) + \sin\frac{\theta}{2} \Gamma_{89\underline{10}} \left( \Gamma_{356} + \Gamma_{0236}\, \gamma \right) \right] \mathcal{F} \epsilon_0$$

$$- M M_{\phi_0\phi_3} M_{\xi_1} \cosh\rho \sinh\rho \sin\beta \left[ \cos\frac{\theta}{2} \left( \Gamma_{256} + \Gamma_{0235}\, \gamma \right) + \sin\frac{\theta}{2} \Gamma_{89} \left( \Gamma_{03} + \Gamma_6\, \gamma \right) \right] \mathcal{F} \epsilon_0$$

$$+ M M_{\phi_0\phi_3} M_{\xi_1}^{-1} \cosh\rho \sinh\rho \sin\beta \left[ \cos\frac{\theta}{2} \left( \Gamma_{03\underline{10}} + \Gamma_{6\underline{10}}\gamma \right) + \sin\frac{\theta}{2} \Gamma_{89\underline{10}} \left( \Gamma_{256} + \Gamma_{0235}\, \gamma \right) \right] \mathcal{F} \epsilon_0 \tag{C.3}$$

The terms in the first 4 lines of the above equation will combine to give the r.h.s. in (C.2) while the terms with $M_{\phi_2\phi_3}$, $M_{\phi_0\phi_2}$ and $M_{\phi_0\phi_3}$ factors will vanish among themselves respectively. Let's focus on them one by one. We first consider $M_{\phi_2\phi_3}$ terms. We impose the projection condition

$$\Gamma_{0235689}\epsilon_0 = \epsilon_0 \tag{C.4}$$

and substitute for $\mathcal{F}$ to get

$$-M M_{\phi_2\phi_3} \cosh^2\rho \cos\beta \sin\beta \left( \cos\frac{\theta}{2} - \sin\frac{\theta}{2} \right) \left[ M_{\xi_1} \left( \Gamma_{026} + \Gamma_{035} \right) - M_{\xi_1}^{-1} \left( \Gamma_{26\underline{10}} + \Gamma_{35\underline{10}} \right) \right] \epsilon_0 \tag{C.5}$$

which vanishes when we substitute our second projection condition

$$(1 + \Gamma_{2536})\,\epsilon_0 \; = \; 0 \tag{C.6}$$

from section 3.2. Next we write the $M_{\phi_0\phi_2}$ terms. After using the two projections mentioned above and substituting for $\mathcal{F}$ we get

$$M\,M_{\phi_0\phi_2}\left(\cos\frac{\theta}{2} - \sin\frac{\theta}{2}\right)\cosh\rho\sinh\rho\cos\beta\left[M_{\xi_1}\left(\Gamma_{356} + \Gamma_{0236}\,\gamma\right) + M_{\xi_1}^{-1}\left(\Gamma_{02\underline{10}} + \Gamma_{5\underline{10}}\,\gamma\right)\right]\epsilon_0 \tag{C.7}$$

the above will vanish if we impose two more projection conditions breaking the supersymmetry to the total factor of $1/16$

$$\Gamma_{025}\,\epsilon_0 \; = \; \epsilon_0\,, \qquad\qquad \Gamma_{789\underline{10}}\epsilon_0 \; = \; \epsilon_0 \tag{C.8}$$

Along the similar lines the terms with $M_{\phi_0\phi_3}$ factor can also be shown to combine and vanish. And after using the same 4 independent projection conditions in (C.4), (C.6), (C.8) one can show that the terms in the first four lines in (C.3) combine to give

$$M\left(M_{\xi_1}\cos\frac{\theta}{2} - M_{\xi_1}^{-1}\Gamma_{89\underline{10}}\sin\frac{\theta}{2}\right)\epsilon_0 \tag{C.9}$$

which is the r.h.s. in $\kappa$ symmetry constraint in (C.2).

# D    $\kappa$-symmetry analysis of the example: $\Phi_2 = 0$

The $\kappa$-symmetry constraint when the fluxes are absent is

$$-\Gamma_{01245\underline{10}}\,\epsilon \; = \; \epsilon \tag{D.1}$$

After commuting $\Gamma_{01245\underline{10}}$ through the exponential factor $M$ in the Killing spinor $\epsilon$ this constraint equation becomes

$$-M\,\Gamma_{01346\underline{10}}\left(\cos\frac{\theta}{2} - \Gamma_{89\underline{10}}\sin\frac{\theta}{2}\right)M_{\xi_1}\,\epsilon_0 \; = \; M\left(\cos\frac{\theta}{2} - \Gamma_{89\underline{10}}\sin\frac{\theta}{2}\right)M_{\xi_1}\,\epsilon_0 \tag{D.2}$$

This holds only at $\theta = \frac{\pi}{2}$, when we impose the projection condition

$$\Gamma_{0134689}\,\epsilon_0 \; = \; \epsilon_0 \tag{D.3}$$

.
**Turning on the new fluxes from section 2.1:**
Next, we consider the flux field $h$ non-zero and take

$$h \; = \; \left(\mathfrak{e}^{014} + \mathfrak{e}^{25\underline{10}}\right)\mathcal{F} \tag{D.4}$$

After some simplifications, the $\kappa$-symmetry constraint condition takes the form given below

$$\left[-\Gamma_{01245\underline{10}} + \left(\Gamma_{014} + \Gamma_{25\underline{10}}\right)\mathcal{F}\right]M\left(\cos\frac{\theta}{2} - \Gamma_{89\underline{10}}\sin\frac{\theta}{2}\right)M_{\xi_1}\,\epsilon_0 \; = \; \epsilon \tag{D.5}$$

Next, we commute all the 6-product and 3-product $\Gamma$ matrices through the exponential factor $M$ in the Killing spinor to write the l.h.s as follows

$$M_{\phi_1\phi_3} \cos\alpha \sin\alpha \, M_{\xi_1} \left[ (\Gamma_{016} + \Gamma_{034}) \cos\frac{\theta}{2} + \Gamma_{89} (\Gamma_{16} + \Gamma_{34}) \sin\frac{\theta}{2} \right] \mathcal{F} \epsilon_0$$

$$-M_{\phi_1\phi_3} \cos\alpha \sin\alpha M_{\xi_1}^{-1} \left[ (\Gamma_{16\underline{10}} + \Gamma_{34\underline{10}}) \cos\frac{\theta}{2} - \Gamma_{89\underline{10}} (\Gamma_{016} + \Gamma_{034}) \sin\frac{\theta}{2} \right] \mathcal{F} \epsilon_0$$

$$+M_{\xi_1} \left[ \Gamma_{0134689} \sin\frac{\theta}{2} + (\Gamma_{014} \cos^2\alpha + \Gamma_{036} \sin^2\alpha) \cos\frac{\theta}{2}\mathcal{F} - \Gamma_{89} (\Gamma_{25} \cos^2\alpha + \Gamma_{14} \sin^2\alpha) \sin\frac{\theta}{2}\mathcal{F} \right] \epsilon_0$$

$$-M_{\xi_1}^{-1} \left[ \Gamma_{0134\underline{610}} \cos\frac{\theta}{2} - (\Gamma_{36\underline{10}} \cos^2\alpha + \Gamma_{14\underline{10}} \sin^2\alpha) \cos\frac{\theta}{2}\mathcal{F} - \Gamma_{89} (\Gamma_{014} \cos^2\alpha + \Gamma_{036} \sin^2\alpha) \sin\frac{\theta}{2}\mathcal{F} \right] \epsilon_0$$

$$\tag{D.6}$$

The r.h.s. in (D.5) is equal to

$$M \left( \cos\frac{\theta}{2} - \Gamma_{89\underline{10}} \sin\frac{\theta}{2} \right) M_{\xi_1} \, \epsilon_0 \, .$$

R.h.s. has no term with a factor $M_{\phi_1\phi_3}$. Therefore the terms in the first two lines in l.h.s. (D.6) must combine and cancel. This will happen only if we impose another projection condition

$$\Gamma_{14} \, \epsilon_0 \, = \, \Gamma_{36} \, \epsilon_0 \tag{D.7}$$

Next, we move to the terms in the last two lines of (D.6), after using the two projection conditions: (D.3), (D.7) and substituting $\mathcal{F} = \frac{\cos\frac{\theta}{2} - \sin\frac{\theta}{2}}{\cos\frac{\theta}{2} + \sin\frac{\theta}{2}}$, terms here combine to give

$$M_{\xi_1} \left[ \sin\frac{\theta}{2} + \left(\cos\frac{\theta}{2} - \sin\frac{\theta}{2}\right) \Gamma_{014} \right] \epsilon_0 \, - \, M_{\xi_1}^{-1} \Gamma_{89\underline{10}} \left[ \cos\frac{\theta}{2} - \left(\cos\frac{\theta}{2} - \sin\frac{\theta}{2}\right) \Gamma_{014} \right] \epsilon_0 \tag{D.8}$$

This will not give the r.h.s. in (D.5) unless we impose one more projection condition

$$\Gamma_{014} \, \epsilon_0 \, = \, \epsilon_0 \tag{D.9}$$

Therefore this brane solution with the flux field value given in (D.4) is 1/8-BPS with the 3-independent projections imposed given below

$$-\Gamma_0 \, \epsilon_0 \, = \, \Gamma_{14} \, \epsilon_0 \, = \, \Gamma_{36} \, \epsilon_0 \, = \, \Gamma_{89} \, \epsilon_0 \tag{D.10}$$

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
