# Peer review of "$\kappa$-symmetric M5 brane web for defects in $AdS_7$ / CFT_6 holography"

_SciPost Physics_

## Round 1 · Referee Report · Anonymous (Referee 1) · 2025-7-12

Strengths

Innovative Extension
Building on the authors’ previous work [9], the paper introduces a general worldvolume self-dual 3-form flux $h$ and provides a complete κ‑symmetry analysis, yielding more general embedding conditions.
Systematic κ‑Symmetry Derivation
Through a detailed expansion of the Γₖ projector operator, the authors derive the corrected embedding equations in the presence of flux (equation (2.25)) and discuss several special cases preserving different amounts of supersymmetry.
Rich Supersymmetric Examples
Section 3 examines a variety of highly supersymmetric examples (½ BPS, ¼ BPS, ⅛ BPS, 1/16 BPS), presenting the corresponding flux configurations and projection conditions.
Physical Insights on Dual Field Theory
The paper analyzes how the flux excitation leads to “ridge-like” deformations of the M5 worldvolume and discusses the anticipated coupling structure of the corresponding boundary defect.
Comprehensive Appendices and References
Appendices A–C supply full details on the background geometry, the six-form constraint, and the κ‑symmetry calculations. The bibliography is thorough and facilitates easy tracking of related work.

Weaknesses

Lack of Visual Illustrations
Figures 1 and 2 are merely conceptual sketches and do not accurately depict the deformed worldvolume geometry in concrete cases.
Insufficient Discussion of the Field Theory Side
Although “boundary defect couplings” are mentioned in Section 4.2, there is no explicit calculation of defect operators or an effective action, leaving the discussion qualitative rather than quantitative.
High Technical Density
The lengthy Γ‑matrix expansions in the κ‑symmetry analysis may be difficult for non‑experts to follow, reducing the paper’s overall readability and flow.
Absence of Numerical or Limiting Analyses
While boundary-limit behavior is discussed qualitatively, there are no concrete expansions or quantitative comparisons for one‑ or two‑point functions of scalar or form‑field operators.

Report

This paper presents a thorough study of κ‑symmetric, flux‑excited M5 probe branes in the AdS₇/CFT₆ duality, generalizing earlier zero‑flux results to derive the more general embedding condition (2.25) and exploring a spectrum of supersymmetric examples from ½ BPS to 1/16 BPS. The derivations are rigorous and complete, and the work provides valuable groundwork for understanding defect–brane duality. However, many key physical conclusions remain at a qualitative level without intuitive figures or quantitative checks on the dual field theory side, which may make it challenging for readers to grasp the geometry of the deformed worldvolume and its precise field‑theoretic implications. Overall, the manuscript offers an important κ‑symmetry framework and new solutions for M5‑brane defects but would benefit from enhanced clarity and concrete physical applications.

Requested changes

Add Illustrative and Numerical Plots
Include cross‑section diagrams of deformed worldvolumes for specific examples (e.g. Φ₁=0, Z₂=0) and, if possible, use simplified numeric interpolation to plot the profile curves.
Supplement Dual Field Theory Calculations
Provide one‑ or two‑point function computations or effective action terms for defect operators to strengthen the physical interpretation.
Streamline Technical Sections
Summarize the main steps of the κ‑symmetry expansion in a table or flowchart, moving the most cumbersome Γ‑matrix details to an appendix.
Quantitative Boundary‑Limit Analysis
Perform explicit expansions as $r\to\infty$, identifying which scalar or two‑form fields exhibit singular behavior, and compare with known results in the literature.
Consolidate Projection Conditions
Present a summary table of the preserved supersymmetries, flux components, and projection operators for the cases in Sections 3.1–3.3, to facilitate quick reference.

Recommendation

Publish (easily meets expectations and criteria for this Journal; among top 50%)

  • validity: high
  • significance: high
  • originality: high
  • clarity: high
  • formatting: excellent
  • grammar: perfect

Author:  Varun Gupta  on 2025-08-06  [id 5703]

(in reply to Report 1 on 2025-07-12)
Category:
answer to question

Dear Editor and Referee,

We would like to thank the referee #1 for a careful reading of the manuscript and the comments. We have taken into account of the suggestions made by the referee in his report and modified the manuscript as follows:

  1. We have added a table in subsection 3.2(table# 1) that describes the intersection of the probe brane worldvolume with the boundary region of $AdS_7$ for the example: $\Phi_1 = 0$; for the instances when the new worldvolume flux is absent and when it is present. Giving a clear idea about the directions in which deformation happens when h is non-zero.
  2. We have moved the details of the kappa symmetry calculation steps for the second example: $\Phi_2 = 0$ in subsection 3.3, to a new appendix D.
  3. We have added a new table at the end of subsection 3.3(table# 2) where the supersymmetry results of all three examples, for all instances of flux field values(presented in subsections 3.1, 3.2, and 3.3), are summarised.

We would like to add that the other two questions raised by the referee regarding the two-point function calculation for defect operators on the field theory side, and a more detailed discussion about the singular behaviour of the bosonic fields at the boundary, are very important. But they fall outside the scope of view with which the current manuscript was written. We will look to reporting on answers to these questions in the near future. We hope these revisions in the manuscript are acceptable to the referee.

---

## Round 1 · Referee Report · Anonymous (Referee 2) · 2025-9-3

Strengths

1.The paper provides a solid extension of previous work on probe M5 branes in AdS$_7 \times$ S$^4$, incorporating worldvolume fluxes into the analysis.
2.The $\kappa$-symmetry analysis is carefully carried out, leading to a classification of supersymmetry-preserving configurations down to $1/16$ BPS.
3.The work provides examples to illustrate the results in a concrete way.

Weaknesses

1.The novelty lies mostly in extending previous results to the flux-turned-on case. 2.The physical interpretation in terms of the dual defect CFT remains less clear, and connections to defect observables could be developed further. 3. The figures are schematic only.

Report

The manuscript presents a careful and detailed analysis of $\kappa$-symmetric probe M5 brane embeddings with worldvolume flux in AdS$_7 \times$ S$^4$. It extends the author’s earlier work by deriving general embedding conditions, classifying supersymmetry breaking patterns, and illustrating the results with examples. While the work may not represent a major breakthrough, it is technically sound, incremental, and of clear relevance to the community working on AdS$_7$/CFT$_6$, defects in SCFTs. Although it lacks of broad cross-field impact, it may fit well within the scope of SciPost Physics Core.

Requested changes

1.Expand the discussion of the dual defect interpretation, including possible connections to displacement operators, or anomaly coefficients.
2.Figures can be improved to be illustrative.
3.Improve the readability by adding short explanatory paragraphs between blocks of algebra, especially in Section 3.

Recommendation

Accept in alternative Journal (see Report)

  • validity: high
  • significance: good
  • originality: high
  • clarity: high
  • formatting: perfect
  • grammar: perfect

Author:  Varun Gupta  on 2025-10-29  [id 5968]

(in reply to Report 2 on 2025-09-03)

Dear Editor and Referee,

We would like to thank the referee for a careful reading of the manuscript and the comments. We have taken into account of the comments made by the referee in his report and made the following revisions in the manuscript:

1) We have expanded subsection 3.5, where we added some more details about the duality with the defects in the boundary gauge theory. Where the dual defects are related to the 'rigid' defects of Gukov-Witten, which preserve an extra R-symmetry. In this subsection, we have also elaborated a bit more on the work to be done in future in order to make connections with relevant fields/operators associated with the dual defects(when $h$ flux field takes new values suggested from section 2).

2) We have added a table(table #1 in section 3.2) in the revised version 2 that complements figure #1 and figure #2, where we indicate the directions along which the worldvolume will be deformed when the flux field 'h' is turned non-zero.

3) In section 3, we have reformatted many equations. We have refined the paragraphs between equations (3.6) and (3.9), highlighting the location of the brane along the $S^4$ polar angular coordinate $\theta$. We have also moved the calculation of example #2($\Phi_2 = 0$; $Z_2 = 0$;) to a new appendix D. We have also added table #2 in this section, where we have collected all the main results of sections 2 and 3 so that a direct comparison can be made between them.

We hope that the manuscript with these revisions is acceptable to the referee for publication.

---

## Round 1 · Referee Report · Anonymous (Referee 3) · 2025-9-29

Strengths

The article performs a comprehensive study of kappa-symmetric M5 branes on AdS7

Weaknesses

  1. Physical implications of long formulas are not discussed.
  2. Several long but unnecessary and poorly formatted formulas are presented.
  3. Sections 2 and 3 are somewhat disconnected, while the latter should be a special case of the former.

Report

This interesting article carries out a detailed study of probe M5 branes in the AdS_4 x S^7 geometry. After performing a general analysis of kappa--symmetric configurations in section 2, the author presents several explicit examples of brane configurations. Curved branes have played an important role in AdS/CFT correspondence, and the current paper makes an incremental technical contribution to understanding of such objects. The derivations are rather straightforward, but physical implications of long formulas are not discussed. To improve readability of this article and to make it suitable for publication in SciPost, I recommend making several changes:

  1. The relevance of the M5 brane configurations for the AdS/CFT correspondence is discussed only in the first paragraph of the introduction. I recommend making at least some minimal comments about dual interpretation of every configuration constructed in this article. This does not have to be an extensive discussion, but the new features of the novel configurations and their field theory duals in comparison to the well-known ones should be stressed.

  2. The paper contains several long equations which are not properly formatted (e.g., (3.3), equations after (3.14), (3.18), (3.21)). I would recommend at least ensuring that this equations have proper margins, but more generally I would question the necessity of giving these long but fairly standard expressions. Projectors for spinors on AdS space are very well known, so I would recommend connecting to the existing literature rather than re-deriving every single projector. Such a change would significantly shorten section three of the article, but it would also separate the novel contribution by the author from quotation of well-known results. If the author feels that full projectors on AdS are needed for completeness, they should be given in appendices, not in the main text. For example, in the main text I would recommend writing just one relevant summarizing relation instead of (3.6)-(3.9).

  3. Currently the examples presented in section 3 are somewhat disconnected from the general discussion given in section 2. I recommend writing a general form of the projector in section 2 in such a way that it can be easily reduced to special cases presented in section 3. This will also address the problem from item 2.

To summarize, this technical article makes incremental contribution to studies of M5 branes in curved backgrounds. While the results might be interesting to researchers working on AdS/CFT correspondence, they are presented in a somewhat convoluted way that mixes the novel contributions with well-known spinor projections. The implications of the new results for the AdS/CFT correspondence are not discussed. I recommend reconsidering this article for publication in SciPost once the major changes outlined above are implemented. I do not recommend publication of this article in its current form.

Requested changes

See items 1-3 in the report.

Recommendation

Ask for major revision

  • validity: good
  • significance: low
  • originality: low
  • clarity: ok
  • formatting: below threshold
  • grammar: excellent

Author:  Varun Gupta  on 2025-10-09  [id 5907]

(in reply to Report 3 on 2025-09-29)
Category:
answer to question

Please see the attached file for the author's response to report #3.

Attachment:

response_letter.pdf

Author:  Varun Gupta  on 2025-10-07  [id 5899]

(in reply to Report 3 on 2025-09-29)
Category:
answer to question

Please see the response letter attached below.

Attachment:

response_letter.pdf

Author:  Varun Gupta  on 2025-10-04  [id 5887]

(in reply to Report 3 on 2025-09-29)

Dear Editor and Referee,

We would like to thank the referee for a careful reading of the manuscript and the comments. We have taken into account the recommendations made by the referee in his report, and we would like to respond in the following paragraphs.

\textbf{Point #1:}

In the submitted version of the manuscript to scipost journal, we had dedicated subsection 3.5 to discuss about the holographic duals of the three probe M5 examples that we discussed in section 3. In this subsection, we focused specifically on the first example $\Phi_1=0;$ $Z_2 = 0;$ to recover the defect in the boundary gauge theory by taking a large value limit of the $AdS_7$ radial coordinate. The steps discussed here also apply to the other two examples to recover the respective defects of the same features and symmetry.

\textbf{Changes made in the revised version:}

We have edited subsection 3.5. We have sharpened the comments made in the discussion in 3.5. We have described the features of the dual defects a bit more and their connection to the \textit{rigid} surface defects of Gukov-Witten in reference [4]. Although the discussion in this section is not a proof of the duality with the \textit{rigid} defects, we have highlighted the features that make the basis of our expectations.

\textbf{Point #2:}

In this manuscript, we have found a new class of M5 brane solutions that preserve at least 1 supersymmetry from the 11d background spacetime geometry of $AdS_7 \times S^4$. There have been many papers in the past that have discussed half-BPS solutions solving the $\kappa$-symmetry constraint equation. But there have been very few papers that have done it for the M5 branes that preserve so little amount of susy. In this article, we work in a special vielbein frame that allows us to express the $AdS_7$ part as a $U(1)$ Hopf fibration over a Kaehler manifold $\widetilde{\mathbb{CP}}^3$, and the $S^3$ inside the $S^4$ part as a $U(1)$ Hopf fibration over a Kaehler manifold $\mathbb{CP}^1$(we had put this detail in appendix A). Due to this choice of frame veilbein, in section 2 of this manuscript, we have been able to derive our general solution for probe M5 in terms of two arbitrary holomorphic conditions in equation (2.25). The analysis that we do in Section 3 is for the simpler case examples coming from the general condition in equation (2.25). In section 3, we have done the $\kappa$-symmetry calculation in detail for the half-BPS example $\Phi_1 = 0;$$Z_2 = 0;$ to emphasise two new features that we noticed in this work:

i) The significance of the position of the probe M5 on the $S^4$ polar coordinate $\theta$ direction. The detailed calculation here shows how the probe M5 is not allowed to carry the new fluxes from section 2(obtained in eqn (2.19)) at $\theta = \pi / 2$ location.

ii) To highlight how the supersymmetry breaks differently when different components of the 3-form flux field $h$ are turned on. For the flux components chosen in subsection 3.2, susy breaks by an additional factor of 1/4. And for the flux components chosen in subsubsection 3.2.2, susy breaks by an additional factor of 1/8.

\textbf{Changes made in the revised version:}

We have reduced the number of steps between equations (3.6) and (3.9). We have only kept two steps before writing the projection condition for the example $\Phi_1 = 0;$$Z_2 = 0;$. The first of those steps is to define the exponential factor $M$ that we use repeatedly in this section. And the second step highlights the location of the brane at $\theta = \pi /2$.
We also restructured many equations in this section, which were written ill-formatted manner previously. We have also relegated the detailed calculation of the example $\Phi_2 = 0;$$Z_2 = 0;$, in subsubsection 3.3.1, to a new appendix D.

\textbf{Point #3:}

\textbf{Changes made in the revised version:}

We have added a new table($\#$2) where we summarise the results of the three half-BPS examples discussed in section 3. In the last two sets of rows of this table, we have also written the summarised results of the general 1/32 BPS solution obtained in section 2, so that a direct comparison can be made between all three examples that descend from the general results of section 2. We have also edited the heading title of subsection 3.2, subsubsections 3.3.1 and 3.3.2, so that their relation with the two conditions in eqn 2.25 becomes more apparent.

We hope these revisions in the manuscript are acceptable to the referee.

(We have attached a pdf file copy of this response, merged with a revised version of the manuscript below.)

Attachment:

response_to_referee.pdf

---

## Round 2 · Referee Report · Anonymous (Referee 3) · 2025-10-24

Strengths

The article performs a comprehensive study of kappa-symmetric M5 branes on AdS7.

Weaknesses

All weaknesses mentioned in the previous report have been addressed.

Report

I thank the author for making the requested changes and for providing additional material that improves readability of this article. I recommend this paper for publication.

Requested changes

None.

Recommendation

Publish (meets expectations and criteria for this Journal)

---

## Round 2 · List of Changes

1. We have added a table in subsection 3.2(table# 1) that describes the intersection of the probe brane worldvolume with the boundary region of $AdS_7$ for the example: $\Phi_1 = 0$; for the instances when the new worldvolume flux is absent and when it is present. Giving a clear idea about the directions in which deformation happens when $h$ is non-zero.

  2. We have edited subsection 3.5. We have sharpened the comments made in the discussion in 3.5. We have described the features of the dual defects a bit more and their connection to the \textit{rigid} surface defects of Gukov-Witten in reference [4]. Although the discussion in this section is not a proof of the duality with the \textit{rigid} defects, we have highlighted the features that make the basis of our expectations.

  3. We have reduced the number of steps between equations (3.6) and (3.9). We have only kept two steps before writing the projection condition for the example $\Phi_1 = 0;$$Z_2 = 0;$. The first of those steps is to define the exponential factor $M$ that we use repeatedly in this section. And the second step highlights the location of the brane at $\theta = \pi /2$. We also restructured many equations in this section, which were written ill-formatted manner previously. We have also relegated the detailed calculation of the example $\Phi_2 = 0;$$Z_2 = 0;$, in subsubsection 3.3.1, to a new appendix D.

  4. We have added a new table($#$2) where we summarise the results of the three half-BPS examples discussed in section 3. In the last two sets of rows of this table, we have also written the summarised results of the general 1/32 BPS solution obtained in section 2, so that a direct comparison can be made between all three examples that descend from the general results of section 2. We have also edited the heading title of subsection 3.2, subsubsections 3.3.1 and 3.3.2, so that their relation with the two conditions in eqn 2.25 becomes more apparent.

---

## Editorial Decision

in_refereeing